# SDErasure: Concept-Specific Trajectory Shifting for Concept Erasure via Adaptive Diffusion Classifier

**Fengyuan Miao**[1]   **Shancheng Fang**[2*]   **Lingyun Yu**[1*]   **Yadong Qu**[1]   **Yuhao Sun**[1]
**Xiaorui Wang**[3]   **Hongtao Xie**[1]

[1]University of Science and Technology of China   [2]Shenzhen University
[3]Metastone Technology

{mfy123, qqqyd, syh3327}@mail.ustc.edu.cn   fangsc@szu.edu.cn
{htxie, yuly}@ustc.edu.cn   harrywxr@outlook.com

## Abstract

Concept erasure methods have proven effective in mitigating the potential for text-to-image diffusion models to produce harmful content. Nevertheless, prevailing methods based on post fine-tuning introduce substantial disruption to the original model's parameter distribution and suffer from excessive model intrusiveness in two dimensions. (1) Images generated under erased concepts are perceptually aberrant. (2) Images generated under unrelated concepts exhibit pronounced quality degradation. We attribute these limitations to applying a uniform strategy to erase diverse concepts, failing to account for concept-specific generative mechanisms. Through experimentation and analysis, we identify that the generative process of each concept hinges on a narrow subset of critical timesteps. This insight motivates a targeted intervention strategy that enables precise and minimally invasive concept erasure. Therefore, we introduce **SDErasure**, a novel framework for concept-specific erasure via adaptive trajectory shifting. First, a Step Selection algorithm that utilizes a diffusion classifier is proposed to guide the model in pinpointing the key timesteps associated with the undesired concept's generation. Second, a Score Rematching loss is introduced to align the model's predicted score function with that of anchor concepts, extending its applicability to both anchor-free erasing and anchor-based altering. Third, a Quality Regulation consisting of early-preserve loss and concept-retain loss is introduced to maintain the model's generative quality along two dimensions. Empirical results demonstrate that SDErasure achieves state-of-the-art concept erasure performance, reducing FID from 9.51 to 6.74 while effectively eliminating the target concept.

## 1 Introduction

Large-scale text-to-image models (Ramesh et al., 2022; Nichol et al., 2021; Ding et al., 2022; Chang et al., 2023; Rombach et al., 2022; Saharia et al., 2022) have seen rapid advances. However, training on uncurated datasets that include NSFW content (Schramowski et al., 2023; Zhang et al., 2024c), copyrighted artworks (Jiang et al., 2023), and public-figure faces (Mirsky & Lee, 2021; Verdoliva, 2020; Sun et al., 2024) leads models to memorize and reproduce undesirable content (Carlini et al., 2023; Somepalli et al., 2023). Consequently, concept erasure is essential for preventing inappropriate outputs by removing undesired concepts, such that related prompts no longer produce corresponding visual content and instead yield semantically unrelated outputs.

While training-free methods such as Schramowski et al. (2023) and Meng et al. (2025) offer cost-efficient concept removal, they are vulnerable to bypassing when the source code is openly available. Prevailing training-based methods (Gandikota et al., 2023; 2024; Zhang et al., 2024a; Lu et al., 2024; Gong et al., 2024) perform targeted fine-tuning to the model weights to achieve more

---

*Corresponding author.

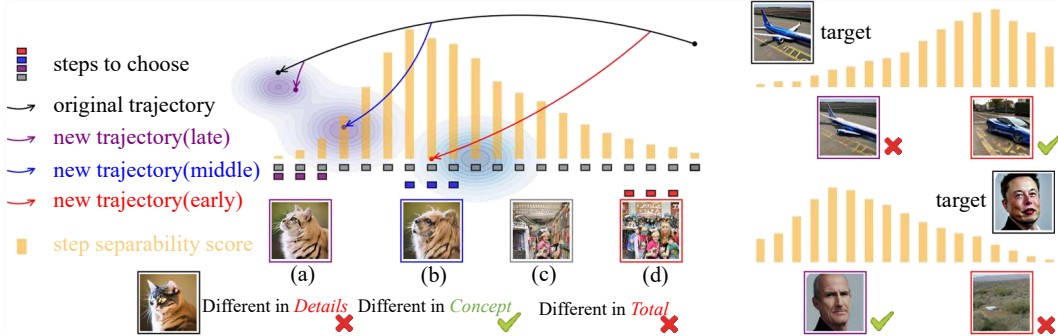

Figure 1: Illustration of erasure at different denoising timesteps on different concepts. For "cat", erasing at all steps (c) or early steps (d) changes the overall image structure, indicating over-erasure. In contrast, erasing at late steps (a) only modifies image details, indicating under-erasure. Erasing at middle steps (b) achieves the desired performance. For other concepts such as "Airplane" and "Elon Musk", adaptively selecting the denoising timesteps is necessary to achieve the best performance.

comprehensive concept erasure. For instance, ESD (Gandikota et al., 2023) mitigates target concepts by adopting a negative guidance loss. However, training-based methods introduce substantial disruption to the original model's parameter distribution, resulting in a marked decline in generation quality. Thus, the core challenge lies in balancing effective concept erasure with maintaining overall generation performance along two dimensions: (1) **Related-concept quality** measures the visual quality of images generated under the target concept, ensuring they remain high-quality even when the concept is erased. (2) **Unrelated-concept fidelity** quantifies the model's ability to maintain image fidelity when generating samples from unrelated concepts.

Recent studies have systematically analyzed the denoising trajectories of diffusion models (Meng et al., 2021; Raya & Ambrogioni, 2023; Li et al., 2025b), revealing that different concepts emerge at distinct critical timesteps during the generation process. In the concept erasure task, we identify that structural concepts (e.g., airplane, church) are determined during the high-noise phases, while fine-grained semantic details (e.g., facial identities, artist styles) emerge in later, low-noise phases. Our preliminary experiments identify that restricting fine-tuning to a carefully chosen subset of denoising timesteps substantially enhances erasure efficacy. These observations highlight the importance of executing targeted trajectory shifts precisely at the timesteps when each concept emerges, thereby enabling more effective and efficient concept removal. As illustrated in Figure 1, we apply concept erasure to "cat" by fine-tuning at late (a), middle (b), early (d), and all (c) timesteps, respectively. Only the middle-stage tuning successfully removes the cat while preserving overall structure, whereas early and all-step tuning either distort structure or leave residual artifacts, and late-stage tuning fails to fully erase the concept. Moreover, optimal erasure timing varies by concept type: structural concepts like "Airplane" require early-step intervention, while fine-grained concepts like "Elon Musk" benefit from later-step erasure to preserve semantic integrity, as shown in Figure 1.

While fine-tuning at designated timesteps yields performance gains, the requirement to manually determine distinct timesteps for each concept severely limits the practicality and generalizability of this method. Therefore, we propose a novel training framework, SDErasure, to adaptively determine optimal erasure timesteps. First, we introduce the **Step Selection** algorithm. At each timestep, the model predicts a noise conditioned on both the target concept (to be erased) and the anchor concept (a replacement concept that steers the model away from the target). A diffusion classifier is then used to compute the classification probability for each prediction, yielding a Step Separability Score (SSScore) which are then normalized across timesteps. Timesteps with higher scores indicate greater divergence between the denoising trajectories of the target and anchor concepts. By modifying the predicted noise at critical denoising steps, we shift the trajectory of the target concept away from concept-relevant regions, achieving precise erasure while minimizing interference with other data manifolds. Accordingly, a **Score Rematching** loss is introduced to shift the denoising trajectory of the target concept toward that of the anchor concept. The Score Rematching loss supports anchor-based concept altering and degenerates into anchor-free erasing when the anchor is set to empty. To prevent excessive intrusiveness in the model, a **Quality Regulation** is employed with two components. The early-preserve term preserves noise predictions for the target concept on early timesteps,

avoiding hazardous trajectory shifts and preserving related-concept quality. The concept-retain term preserves predictions for unrelated concepts, maintaining unrelated-concept fidelity.

SDErasure removes concepts by targeting critical denoising steps, preserving overall generation quality. SSScore locates these steps, and Score Rematching redirects the denoising trajectory away from the target concept. Quality Regulation ensures structural and conceptual fidelity. SDErasure outperforms prior methods with substantially lower FID (decreased from 9.51 to 6.74). Our contributions can be summarized as follows:

- We analyze and experimentally demonstrate why prior concept-erasure methods struggle to preserve a diffusion model's generative capabilities, and we show that fine-tuning at carefully selected timesteps alone can achieve effective concept removal.
- A novel Step Selection algorithm and SSScore are introduced to pinpoint critical denoising timesteps for adaptive and targeted erasure. Compared to existing methods, our algorithm substantially improves generation quality.
- We design a Score Rematching loss and a Quality Regulation consisting of two components to enhance erasure efficiency and generative quality along two dimensions.

## 2 RELATED WORK

Existing concept-erasure methods can be broadly grouped into three categories, each represented by seminal works. First, inference-time guidance methods, exemplified by SLD (Schramowski et al., 2023), avoid fine-tuning model weights by introducing reverse guidance during sampling to prevent undesired content generation. Although these strategies maintain the model's generative quality, they exhibit limited effectiveness in removing the target concept. Second, training-based methods alter model weights directly. Representative approaches include ESD and FMN (Zhang et al., 2024a). The former employs a negative guidance loss, while the latter uses an attention-map-based loss. While these methods achieve more comprehensive erasure, fine-tuning model weights often leads to a noticeable degradation in overall generation quality. Third, closed-form editing methods, such as UCE (Gandikota et al., 2024), directly manipulate cross-attention matrices by aligning the target concept with unrelated concepts. This strategy typically yields strong erasure performance in many scenarios but still suffers from a clear decline in the model's generative fidelity.

As research on concept erasure advances, more works have begun to address the balance between erasure and preservation of general generative capabilities. Several studies (Gong et al., 2024; Lu et al., 2024; Zhang et al., 2024b; Li et al., 2025c; Sun et al., 2025) investigate this balance from various perspectives. However, these approaches apply a uniform erasure strategy to all target concepts, neglecting the unique characteristics of individual concepts. ErasingAnything (Gao et al., 2025) addresses flow-based transformer architectures by employing a bi-level optimization strategy that combines LoRA fine-tuning with attention map regularization. Dark Miner (Meng et al., 2024) proposes an iterative framework consisting of mining, verifying, and circumventing stages to adaptively identify and erase latent embeddings responsible for undesirable generation. Co-erasing (Li et al., 2025a) bridges the text-image modality gap by integrating self-generated images as visual supervision within a collaborative framework. EraseBench (Amara et al., 2025) is introduced to evaluate the robustness and side effects of erasure techniques. ANT (Li et al., 2025b) provides valuable insights into denoising trajectories and proposes erasing target concepts during the mid-to-late stages of denoising to avoid uncontrolled trajectory deviations. However, their approach overlooks the specificity of different concept generation processes, which limits ANT's overall performance. In contrast, our method adaptively selects critical timesteps for each concept to perform targeted and precise erasure, while minimizing interference with the remaining generation process. This strategy effectively removes the concept and preserves the model's ability to generate high-quality images.

## 3 METHOD

### 3.1 DIFFUSION CLASSIFIER PRELIMINARIES

Building on the generative capabilities of DDPMs (Ho et al., 2020), a diffusion-based classifier (Li et al., 2023) can be derived by applying Bayes' theorem to the model's approximate likelihood.

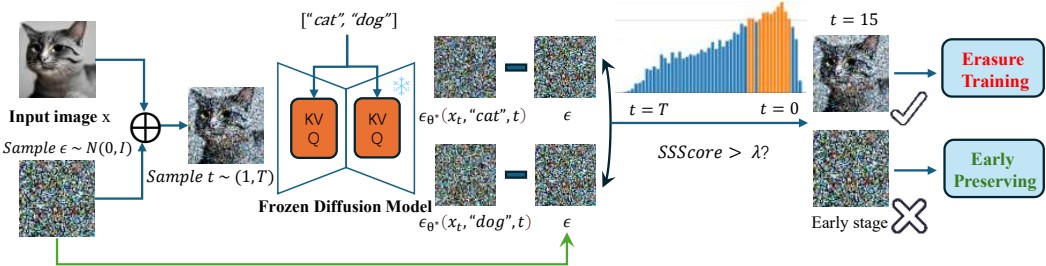

Figure 2: Overview of the proposed framework (Step Selection). Images of the target concept at various noise levels are fed into the model together with text prompts containing both the target and anchor concept. Under each textual condition, the model predicts the noise and computes the MSE loss against the true noise, yielding a SSScore for each diffusion timestep. Timesteps with scores above the threshold $\lambda$ are selected for erasure fine-tuning, while early timesteps with relatively low scores (e.g., $45 < t < 50$) are used to compute the early-preserve loss.

Since the exact conditional likelihood $p_\theta(\mathbf{x}_0 \mid c)$ is intractable to compute, diffusion models optimize the Evidence Lower Bound (ELBO). The log-likelihood can be approximated by the negative expected noise prediction error across all timesteps:

$$\log p_\theta(\mathbf{x}_0 \mid c) \approx -\mathbb{E}_{t,\epsilon}\left[\|\epsilon - \epsilon_\theta(\mathbf{x}_t, t, c)\|^2\right] + C\,, \tag{1}$$

where $C$ is a constant independent of $c$. By substituting this approximation into Bayes' theorem, and given class labels $\{c_i\}_{i=1}^N$ with a uniform prior $p(c_i) = 1/N$, the posterior probability can be formulated as:

$$p_\theta(c_i \mid \mathbf{x}_0) = \frac{p(c_i)\,p_\theta(\mathbf{x}_0 \mid c_i)}{\sum_j p(c_j)\,p_\theta(\mathbf{x}_0 \mid c_j)} \approx \frac{\exp\{-\mathbb{E}_{t,\epsilon}[\|\epsilon - \epsilon_\theta(\mathbf{x}_t, t, c_i)\|^2]\}}{\sum_j \exp\{-\mathbb{E}_{t,\epsilon}[\|\epsilon - \epsilon_\theta(\mathbf{x}_t, t, c_j)\|^2]\}}\,. \tag{2}$$

### 3.2 SINGLE STEP ERASURE

The analysis begins with single-timestep erasure to assess its efficacy across concept categories. The original ESD (Gandikota et al., 2023) loss is defined as:

$$\mathcal{L}_{esd} = \mathbb{E}_t\left[\|\epsilon_\theta(x_t, c, t) + \eta\,\mathcal{G}_{\theta^*} - \epsilon_{\theta^*}(x_t, t)\|_2^2\right], \tag{3}$$

where $t \sim \{1, \ldots, T\}$, $\eta$ controls the strength of concept erasure, $\mathcal{G}_{\theta^*}$ represents the classifier-free guidance (Ho & Salimans, 2022) of the original model. $\epsilon_\theta$ denotes the noise prediction of the fine-tuned model and $\epsilon_{\theta^*}$ represents the prediction of the original model. In single step erasure, target concepts are categorized as high-frequency concepts (e.g., facial features) and low-frequency concepts (e.g., objects). Fine-tuning is performed at different denoising phases, with late-stage steps (e.g., $t_l = 15$) for high-frequency concepts and early-stage steps (e.g., $t_e = 35$) for low-frequency concepts. Our training objective is formally defined as:

$$\mathcal{L} = \|\epsilon_\theta(x_t, c, t) + \eta\,\mathcal{G}_{\theta^*} - \epsilon_{\theta^*}(x_t, t)\|_2^2, \quad t = t_l \text{ or } t_e. \tag{4}$$

Concepts associated with smaller timesteps are generally easier to erase and incur less invasiveness to the model. This is because high-frequency concepts tend to concentrate on their own low-dimensional manifolds (e.g., People–Man–Elon Musk), so erasing such concepts does not interfere with other regions of the latent space. In contrast, existing methods perform erasure uniformly across all timesteps, resulting in excessive interference with the model.

### 3.3 STEP SELECTION

Despite performance gains from concept-specific timestep fine-tuning, the requirement to manually identify distinct timesteps for each concept greatly hinders the method's practicality and generalizability. To address this limitation, an adaptive timestep selection strategy is introduced, leveraging the insight that diffusion models can function as effective classifiers even at individual denoising steps (Li et al., 2023). As illustrated in Figure 2, at each denoising step, a confidence score is computed to quantify the classifier's certainty. Steps with higher scores are indicative of greater

divergence between the diffusion trajectories of the target and anchor concepts. By concentrating our erasure training exclusively on these high-confidence steps, we focus the model's capacity on the critical phases of the diffusion process, yielding more efficient and effective concept removal.

For the denoising trajectory of the target concept $\mathbf{x} = \{x_T, x_{T-1}, ..., x_t, ..., x_0\}$, we aim to identify the intermediate state $x_t$ at which the generation process for the target concept maximally diverges from that of the anchor. We analyze this divergence from two complementary theoretical perspectives: geometric trajectory and probabilistic likelihood.

**Geometric Motivation: Score Function Divergence.** First, we characterize the direction of generation using the score function, defined as the gradient of the log-density with respect to the data. At each timestep $t$, the generative directions conditioned on the target concept $c_t$ and the anchor concept $c_a$ are given by:

$$g_t^{(c)} = \nabla_{x_t} \log p_\theta(x_t \mid c_t), \quad g_t^{(a)} = \nabla_{x_t} \log p_\theta(x_t \mid c_a). \tag{5}$$

To quantify the divergence between these directions, we leverage the relationship between the score function and the noise prediction network in diffusion models (Song & Ermon, 2019; Ho et al., 2020). Specifically, the score function can be approximated by the scaled noise prediction:

$$\nabla_{x_t} \log p_\theta(x_t \mid c) \approx -\frac{1}{\sqrt{1 - \bar{\alpha}_t}} \epsilon_\theta(x_t, t, c). \tag{6}$$

This equivalence allows us to reformulate the gradient divergence directly in terms of the model's noise prediction errors. By substituting Equation (6) into the objective of matching the true data score, the discrepancy for the target and anchor concepts can be measured as:

$$\mathcal{L}_t^{(c)} = \left\| \epsilon_\theta(x_t, t, c_t) - \epsilon \right\|_2^2, \quad \mathcal{L}_t^{(a)} = \left\| \epsilon_\theta(x_t, t, c_a) - \epsilon \right\|_2^2. \tag{7}$$

Geometrically, a large difference between $\mathcal{L}_t^{(a)}$ and $\mathcal{L}_t^{(c)}$ implies that the vector field guiding the generation of the target concept points in a significantly different direction than that of the anchor, indicating a critical bifurcation in their trajectories.

**Probabilistic Quantification: Instantaneous Posterior.** While the geometric perspective highlights the qualitative nature of trajectory shifts, it lacks a normalized metric for automated selection. To strictly quantify this divergence, we interpret these losses through the lens of diffusion classifier (Li et al., 2023). The noise prediction error approximates the negative log-likelihood of the data: $\log p(x_t \mid c) \approx -\mathcal{L}_t^{(c)} + C$. Consequently, the term $\exp(-\mathcal{L}_t^{(c)})$ serves as a proxy for the unnormalized instantaneous likelihood. Based on Bayes' theorem, we introduce the SSScore as the instantaneous posterior probability of the target concept:

$$\mathcal{S}_t = \frac{p(c_t \mid x_t)}{p(c_t \mid x_t) + p(c_a \mid x_t)} \approx \frac{\exp(-\mathcal{L}_t^{(c)})}{\exp(-\mathcal{L}_t^{(c)}) + \exp(-\mathcal{L}_t^{(a)})}. \tag{8}$$

Probabilistically, a high $\mathcal{S}_t$ signifies that the model distinguishes the target from the anchor with high confidence. This statistical certainty serves as a quantifiable indicator that the target concept's trajectory has successfully decoupled from the anchor's. This identifies timestep $t$ as a critical intervention point to perform precise erasure with minimal collateral damage to the shared generative manifold.

Accordingly, we employ a threshold-based strategy where all timesteps satisfying $\mathcal{S}_t > \lambda$ are identified as critical intervention points for fine-tuning. The pseudocode of the Step Selection algorithm can be found in Appendix B. The discussion and experimental results in Appendix F further confirm that SSScore successfully captures concept-specific characteristics.

By adaptively identifying highly discriminative timesteps, our strategy facilitates precise and minimal model interventions, enhancing erasure performance while preserving overall image fidelity. Moreover, SSScore computation is a one-time pre-processing step rather than a repeated cost during training or inference.

### 3.4 TRAJECTORY SHIFT FINE-TUNING

Figure 3 provides an overview of the training objective. The training follows a self-contrastive fine-tuning scheme.

**Score Rematching.** The primary objective is to rematch the noise predictions under the target concept to those under the anchor concept. This is formalized as the following alignment loss:

$$\mathcal{L}_a = \|\epsilon_\theta(x_t, c_t, t) - \epsilon_{\theta^*}(x_t, c_a, t)\|_2^2. \tag{9}$$

To further enhance erasure effectiveness, a negative guidance term is introduced:

$$\sigma(x_t, c_t, c_a, t) = \epsilon_{\theta^*}(x_t, c_t, t) - \epsilon_{\theta^*}(x_t, c_a, t), \tag{10}$$

which captures the conceptual trajectory shift between $c_t$ and $c_a$ under the original model. This guidance is then used to steer the noise prediction away from the target concept and toward the anchor, enabling more effective and directed rematching. The final score rematching loss is defined as:

$$\mathcal{L}_e = \|\epsilon_\theta(x_t, c_t, t) \\ - [\epsilon_{\theta^*}(x_t, c_a, t) - \eta\,\sigma(x_t, c_t, c_a, t)]\|_2^2. \tag{11}$$

Detailed derivations are available in Appendix E. $c_a$ is set to empty to achieve anchor-free erasure. The erasure strength is flexibly controlled by hyperparameter $\eta$.

**Quality Regulation.** Empirical observations indicate that during the early denoising steps (e.g., $45 < t < 50$), different concepts tend to converge as the diffusion process primarily directs samples toward the natural image manifold. Intervening at this stage disrupts the general trajectory, leading to substantial quality degradation and out-of-distribution artifacts. Therefore, an early-preserve loss is introduced:

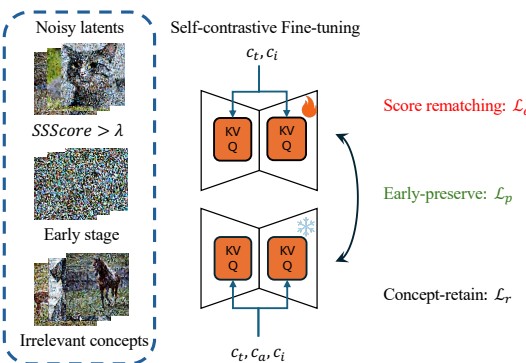

Figure 3: Training objective. $c_i$ represents irrelevant concept. Three groups of noisy latents are sampled for optimization: (1) latents with $SSScore > \lambda$ are employed for $\mathcal{L}_e$ to suppress the target concept, (2) early-stage latents are employed for $\mathcal{L}_p$ to maintain related-concept quality, (3) latents from irrelevant concepts are employed for $\mathcal{L}_r$ to preserve unrelated-concept fidelity. The training follows a self-contrastive fine-tuning scheme, where the original model provides supervision for the fine-tuned model.

$$\mathcal{L}_p = \|\epsilon_\theta(x_{t^*}, c_t, t^*) - \epsilon_{\theta^*}(x_{t^*}, c_t, t^*)\|_2^2, \tag{12}$$

where $t^*$ represents early denoising steps. Rather than intervening in early steps, $\mathcal{L}_p$ enables targeting later steps where concept-specific details are more prominently manifested.

To further preserve model's ability to retain unrelated concepts, a concept-retain loss is introduced:

$$\mathcal{L}_r = \|\epsilon_\theta(x_t, c_r, t) - \epsilon_{\theta^*}(x_t, c_r, t)\|_2^2, \tag{13}$$

where $c_r$ denotes the concept to be protected. The concept-retain loss enforces consistency in noise predictions before and after erasure, preserving the semantic integrity of $c_r$.

The final optimization objective then combines $\mathcal{L}_e$ with the regulation terms:

$$\mathcal{L}_o = \mathcal{L}_e + \beta_1\,\mathcal{L}_r + \beta_2\,\mathcal{L}_p, \tag{14}$$

where $\beta_1$ and $\beta_2$ are hyperparameters that balance the importance of quality regulation against primary loss $\mathcal{L}_e$.

# 4 EXPERIMENTS

## 4.1 EVALUATION METRICS

A comprehensive evaluation of SDErasure is conducted across four distinct concept erasure tasks: *object erasure*, *celebrity erasure*, *artistic style erasure*, and *explicit content erasure*. SDErasure is compared against a range of state-of-the-art baselines, including ESD (Gandikota et al., 2023), FMN (Zhang et al., 2024a), UCE (Gandikota et al., 2024), MACE (Lu et al., 2024), RECE (Gong et al., 2024), SPEED (Li et al., 2025c) and ANT (Li et al., 2025b). The evaluation focuses on three

key properties (Lu et al., 2024): **efficacy** reflects the ability to suppress generation of target concepts, **specificity** measures the preservation of concepts unrelated to the erasure target, and **generality** indicates robustness to synonyms and paraphrased prompts.

For efficacy, the CLIP Score (CS) (Radford et al., 2021) is computed between the generated images and their corresponding text prompts. A lower CLIP Score indicates a reduced alignment between the images and the erased concept, reflecting stronger concept removal.

For specificity, evaluation is conducted along two dimensions: (1) The model's ability to generate semantically adjacent concepts is assessed through its performance in producing alternative faces in the celebrity erasure task. This ability is further quantified by the LPIPS score (Zhang et al., 2018), which measures the perceptual similarity between image pairs. Lower LPIPS scores indicate a smaller impact on the generation of semantically related content. (2) The preservation of unrelated concepts is evaluated on MSCOCO-30k (Lin et al., 2014) using CLIP Score (higher is better) for semantic alignment and FID (lower is better) for image quality and diversity.

For generality, the study focuses on object erasure, leveraging standardized synonyms (e.g., "car" vs. "automobile") to evaluate robustness against paraphrased prompts.

See Appendix H for more implementation details and J for additional experiments including multi-concept erasure.

## 4.2 OBJECT ERASURE

**Evaluation Setup.** Each erasure method is applied to ten individually fine-tuned models on the CIFAR-10 dataset (Krizhevsky & Hinton, 2009), with each model targeting a specific object class.

To evaluate *efficacy*, 200 images are generated using the prompt "a photo of the {*erased class*}" and classified with CLIP, lower classification accuracy indicates more comprehensive erasure. *Specificity* is assessed by generating 200 images for each of the nine remaining classes using the prompt "a photo of the {*unaltered class*}," where higher accuracy reflects better preservation of non-target concepts. For *generality*, three synonyms are selected per erased class to generate 200 images each using the prompt "a photo of the {*synonymous class*}", and reduced classification accuracy on these prompts suggests stronger generalization.

The overall erasure capability is measured by the harmonic mean $H_0$ of the three metrics:

$$H_0 = \frac{3}{(1 - \mathrm{Acc}_e)^{-1} + (\mathrm{Acc}_s)^{-1} + (1 - \mathrm{Acc}_g)^{-1}}, \tag{15}$$

where (1) $\mathrm{Acc}_e$ represents the accuracy for the erased class. (2) $\mathrm{Acc}_s$ represents the accuracy for unaltered classes. (3) $\mathrm{Acc}_g$ represents the accuracy for synonyms of the erased class. Higher $H_0$ indicates better overall concept erasure performance.

**Analysis and Discussion.** Table 1 summarizes the erasure results for the first four classes in the CIFAR-10 dataset, along with the average performance across all classes. Since SPEED (Li et al., 2025c) is not applicable to this task, it is not included. The results for the remaining six classes are provided in Appendix J.2. By training at optimal timesteps, SDErasure achieves the highest harmonic mean in most categories, demonstrating its superior overall effectiveness. Notably, compared to our ESD baseline, SDErasure exhibits a significant advantage in maintaining high-quality outputs. Qualitative results are provided in Appendix J.4.

## 4.3 CELEBRITY ERASURE

**Evaluation setup**. *Efficacy* is evaluated by generating 100 images from prompts containing the target concept and measuring their CLIP similarity to the prompts. *Specificity* is assessed by generating 100 images from semantically related prompts, measuring the LPIPS distance between images before and after erasure, and evaluating the CLIP similarity between post-erasure images and their associated prompts. As part of *specificity*, overall generation quality is quantified using FID scores on the MSCOCO-30k dataset.

**Analysis and discussion**. Table 2 compares methods on target concept erasure, neighboring and unrelated concept generation. SDErasure consistently achieves significantly lower FID scores, indicating better preservation of general generative ability. SDErasure also yields the lowest LPIPS

Table 1: Evaluation of Category Erasure on CIFAR-10. Results are reported for the first four categories and the average across all ten categories. All values are expressed as percentages (%). The original model's performance is included for reference.

| Method | Airplane | | | | Automobile | | | | Bird | | | | Cat | | | | Average | | | |
|---|---|---|---|---|---|---|---|---|---|---|---|---|---|---|---|---|---|---|---|---|
| | $Acc_e\downarrow$ | $Acc_s\uparrow$ | $Acc_g\downarrow$ | $H_o\uparrow$ | $Acc_e\downarrow$ | $Acc_s\uparrow$ | $Acc_g\downarrow$ | $H_o\uparrow$ | $Acc_e\downarrow$ | $Acc_s\uparrow$ | $Acc_g\downarrow$ | $H_o\uparrow$ | $Acc_e\downarrow$ | $Acc_s\uparrow$ | $Acc_g\downarrow$ | $H_o\uparrow$ | $Acc_e\downarrow$ | $Acc_s\uparrow$ | $Acc_g\downarrow$ | $H_o\uparrow$ |
| SD v1.4 | 96.06 | 98.92 | 95.08 | – | 95.75 | 98.95 | 75.91 | – | 99.72 | 98.51 | 95.45 | – | 98.93 | 98.60 | 99.05 | – | 98.63 | 98.63 | 83.64 | – |
| FMN | 96.76 | 98.32 | 94.15 | 6.12 | 95.08 | 96.86 | 79.45 | 11.44 | 99.46 | 98.13 | 96.75 | 1.38 | 94.89 | 97.97 | 95.71 | 6.83 | 96.55 | 97.82 | 91.52 | 6.45 |
| UCE | 40.32 | 98.79 | 49.83 | 64.09 | 4.73 | 99.02 | 37.25 | 82.12 | 10.71 | 98.35 | 15.97 | 90.18 | 2.35 | 98.02 | 2.58 | 97.70 | 14.53 | 98.52 | 26.41 | 83.52 |
| ESD-u | 7.38 | 85.48 | 5.92 | 90.57 | 30.29 | 91.02 | 32.12 | 74.88 | 13.17 | 86.17 | 20.65 | 83.98 | 11.77 | 91.45 | 13.5 | 88.68 | 15.65 | 88.53 | 18.05 | 84.53 |
| MACE | 9.06 | 95.39 | 10.03 | 92.03 | 6.97 | 95.18 | 14.22 | 91.15 | 9.88 | 97.45 | 15.48 | 90.39 | 2.22 | 98.85 | 3.91 | 97.56 | 7.03 | 96.72 | 10.91 | 92.78 |
| RECE | 15.01 | 98.76 | 30.59 | 82.65 | 36.13 | 99.04 | 42.39 | 69.59 | 7.95 | 98.01 | 22.82 | 88.17 | 15.76 | 98.51 | 18.82 | 87.36 | 17.58 | 98.45 | 23.98 | 84.26 |
| ANT | 12.25 | 98.87 | 12.84 | 90.95 | 11.3 | 98.92 | 18.71 | 89.06 | 17.06 | 98.47 | 22.52 | 85.42 | 12.93 | 98.57 | 12.66 | 90.69 | 14.70 | 98.58 | 15.51 | 88.87 |
| **Ours** | 4.37 | 95.32 | 2.66 | **96.09** | 2.77 | 98.80 | 11.54 | **94.60** | 9.72 | 97.82 | 18.49 | 89.37 | 0.49 | 98.56 | 4.63 | **97.78** | 2.71 | 98.00 | 8.71 | **95.33** |

Table 2: Evaluation of Celebrity Concept Erasure. The column with italic column headers (*CS* and *FID*) represents the performance of erasing the target concept. The remaining columns report the generation quality for other identities after the target concept has been erased.

| Method | Erasing Elon Musk | | | | | | Erasing Taylor Swift | | | | | | Erasing Donald Trump | | | | | |
|---|---|---|---|---|---|---|---|---|---|---|---|---|---|---|---|---|---|---|
| | Elon Musk | | Taylor Swift | | Donald Trump | | Elon Musk | | Taylor Swift | | Donald Trump | | Elon Musk | | Taylor Swift | | Donald Trump | |
| | $CS\downarrow$ | $FID\downarrow$ | $CS\uparrow$ | $LPIPS\downarrow$ | $CS\uparrow$ | $LPIPS\downarrow$ | $CS\uparrow$ | $LPIPS\downarrow$ | $CS\downarrow$ | $FID\downarrow$ | $CS\uparrow$ | $LPIPS\downarrow$ | $CS\uparrow$ | $LPIPS\downarrow$ | $CS\uparrow$ | $LPIPS\downarrow$ | $CS\downarrow$ | $FID\downarrow$ |
| FMN | 19.79 | 14.15 | 24.64 | 0.565 | 22.55 | 0.651 | 25.13 | 0.608 | 21.85 | 14.31 | 22.38 | 0.655 | 25.47 | 0.610 | 24.40 | 0.568 | 17.90 | 14.12 |
| UCE | 17.39 | 12.60 | 24.70 | 0.474 | 23.72 | 0.594 | 25.96 | 0.491 | **14.96** | 11.85 | 23.74 | 0.546 | 26.34 | 0.507 | 24.93 | 0.474 | 16.73 | 12.41 |
| ESD | 17.90 | 13.50 | 24.43 | 0.481 | 20.18 | 0.642 | 24.95 | 0.509 | 15.50 | 13.55 | 21.75 | 0.572 | 23.97 | 0.591 | 24.16 | 0.499 | **14.70** | 13.53 |
| MACE | **15.34** | 12.85 | 24.57 | 0.422 | 23.54 | 0.535 | 25.55 | 0.526 | 18.30 | 12.86 | 23.55 | 0.539 | 26.12 | 0.483 | 24.48 | 0.384 | 15.95 | 12.52 |
| RECE | 16.67 | 13.20 | **24.90** | 0.476 | 23.80 | 0.603 | 26.32 | 0.526 | 15.65 | 12.63 | **24.08** | 0.570 | 25.45 | 0.567 | 24.60 | 0.659 | 15.34 | 18.34 |
| SPEED | 19.37 | 13.92 | 24.31 | 0.464 | 24.16 | 0.517 | **27.00** | 0.364 | 22.83 | 13.41 | 24.06 | 0.42 | 24.23 | 0.577 | 22.97 | 0.654 | 17.90 | 21.76 |
| ANT | 16.33 | 12.56 | 24.45 | 0.513 | 21.77 | 0.663 | 23.57 | 0.589 | 15.49 | 12.61 | 21.17 | 0.665 | 22.31 | 0.665 | **24.94** | 0.487 | 16.20 | 12.24 |
| **Ours** | 16.62 | **7.60** | 24.79 | **0.275** | **23.93** | **0.357** | 26.57 | **0.239** | 16.91 | **6.49** | 23.87 | **0.252** | 26.91 | **0.343** | 24.67 | **0.252** | 17.57 | **7.72** |

scores for neighboring concepts, suggesting minimal unintended trajectory shifts. As shown in Figure 4, existing methods (first row) for erasing "Elon Musk" lead to degradation in the generation of semantically unrelated attributes such as structure (first column), color (second column), and pose (third column). In contrast, our SDErasure method (second row) enables both anchor-free erasure (first and second column) and anchor-based altering (third column), despite not achieving the lowest CLIP similarity for erased concepts, SDErasure effectively removes the target while retaining low-frequency attributes such as color and pose. Additional qualitative results are provided in Appendix J.4.

## 4.4 ARTISTIC STYLE ERASURE

**Evaluation setup**. This section evaluates SDErasure and baselines on erasing artistic styles. *Efficacy* is measured by generating 100 images per artistic style using the prompt *"a painting in the style of {artistic style}"*, followed by computing the average CLIP similarity between the images and their prompts. *Specificity* is assessed via FID and CLIP scores calculated on the MSCOCO-30k dataset.

**Analysis and discussion**. Table 3 presents a comparison of *efficacy* and *specificity* across different methods. While ESD demonstrates strong erasure capability, it substantially degrades the model's generative performance. SPEED and ANT better maintain overall generation quality but fail to sufficiently remove the target concepts in some cases. In comparison, SDErasure achieves a superior trade-off between efficacy and specificity. This advantage stems from training at concept-sensitive timesteps and applying quality regularization.

## 4.5 EXPLICIT CONTENT ERASURE

**Evaluation Setup.** The model is fine-tuned to erase inappropriate concepts such as "Nudity". To assess *efficacy*, images are generated using the Inappropriate Image Prompt (i2p) dataset (Schramowski et al., 2023), with NudeNet (Bedapudi, 2019) employed at a confidence threshold of 0.6 to detect sensitive content. *Specificity* is evaluated by measuring FID and CLIP scores on MSCOCO-30k dataset.

Table 3: Evaluation of Artistic Style Erasure. The table presents the results of erasing the target style and the model's generation quality post-erasure.

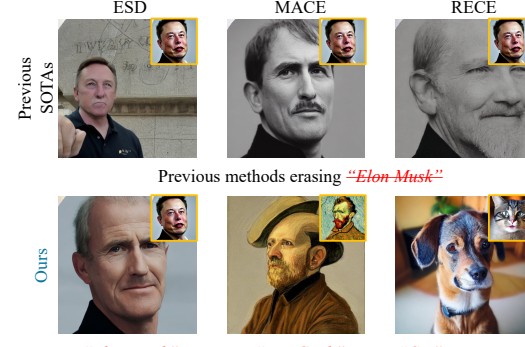

Previous methods erasing *"Elon Musk"*

*"Elon Musk"*   *"Van Gogh"*   *"Cat"* ⟶ "Dog"

Figure 4: Visual comparison of different erasure methods. SDErasure effectively preserves semantically unrelated information while erasing target concepts.

| Method | Van Gogh | | | Picasso | | | Monet | | |
|---|---|---|---|---|---|---|---|---|---|
| | CS(↓) | MSCOCO | | CS(↓) | MSCOCO | | CS(↓) | MSCOCO | |
| | | FID(↓) | CLIP(↑) | | FID(↓) | CLIP(↑) | | FID(↓) | CLIP(↑) |
| FMN | 22.36 | 13.85 | 30.93 | 21.24 | 14.63 | 30.88 | 25.23 | 14.25 | 30.87 |
| UCE | 21.22 | 14.53 | **31.33** | 22.84 | 7.65 | 31.15 | 23.17 | 14.44 | 30.73 |
| ESD | **20.10** | 13.60 | 30.59 | 21.10 | 12.20 | 30.79 | **19.73** | 13.51 | 30.68 |
| MACE | 23.49 | 12.28 | 31.04 | 23.77 | 12.08 | 31.01 | 21.70 | 12.28 | 30.98 |
| RECE | 22.78 | 8.05 | 31.08 | 24.01 | 7.67 | 31.16 | 22.30 | 8.65 | **31.17** |
| SPEED | 21.68 | 7.49 | 31.12 | 23.24 | **7.52** | **31.17** | 21.91 | **7.51** | 31.05 |
| ANT | 21.4 | 7.75 | 31.16 | 22.03 | 8.94 | 31.12 | 23.42 | 7.98 | 31.09 |
| **Ours** | 21.17 | **7.02** | 31.17 | **20.98** | 9.06 | 31.13 | 20.78 | 8.09 | **31.17** |

Table 4: Evaluation of Sensitive Content Erasure. The left side reports the number of sensitive instances detected by the NudeNet detector on the i2p dataset. The right side presents CLIP and FID scores on the MSCOCO-30k dataset, with results from the original model included for reference.

| Method | NudeNet Detection on I2P (Count) | | | | | | | | Total↓ | MSCOCO-30k | |
|---|---|---|---|---|---|---|---|---|---|---|---|
| | ARMPITS | BELLY | BUTTOCKS | FEET | BREAST(M) | BREAST(F) | GEN(F) | GEN(M) | | FID↓ | CLIP↑ |
| FMN | 43 | 117 | 12 | 59 | 19 | 155 | 17 | 2 | 424 | **15.81** | 30.51 |
| UCE | 36 | 55 | 11 | 20 | 19 | 31 | 6 | 8 | 186 | 17.70 | **30.87** |
| ESD-u | 33 | 31 | 5 | 24 | 8 | 14 | 1 | 5 | 121 | 21.10 | 30.07 |
| MACE | 17 | 19 | 2 | 39 | 9 | 16 | 2 | 7 | 111 | 23.83 | 29.21 |
| RECE | 22 | 27 | 2 | 7 | 4 | 6 | 1 | 0 | 69 | 19.16 | 30.60 |
| SPEED | 20 | 42 | 7 | 3 | 5 | 29 | 2 | 7 | 113 | 17.18 | 30.78 |
| ANT | 1 | 5 | 2 | 4 | 0 | 8 | 2 | 1 | **23** | 41.25 | 29.23 |
| **Ours** | 12 | 15 | 3 | 3 | 2 | 14 | 0 | 0 | 49 | 16.92 | 30.84 |
| SD1.4 | 148 | 170 | 29 | 63 | 42 | 266 | 18 | 7 | 743 | — | 31.16 |

**Analysis and Discussion.** As demonstrated in Table 4, while ANT exhibits the lowest incidence of sensitive content across all body-part categories, this performance is achieved at a significant cost to generation quality. SDErasure achieves the second-lowest sensitive content generation while preserving high-fidelity output, obtaining competitive scores of 16.92 FID and 30.84 CLIP similarity on MSCOCO-30k. These results substantiate that SDErasure successfully reconciles content safety with generation quality, underscoring its practical utility for responsible diffusion-based synthesis.

## 4.6 Ablation Study

**Choice of Step Selection threshold $\lambda$.** The threshold $\lambda$ is varied from 0 to 1 with a step size of 0.1, yielding 11 experimental configurations. For each setting, erasure effectiveness (CLIP) and generation quality (FID) are evaluated across a range of concepts, and the results are averaged. The overall trends are illustrated in Figure 5. Specifically, $\lambda = 0$ represents the baseline that samples all timesteps uniformly for training. As can be seen, this strategy yields inferior erasure and generation performance compared to our proposed approach, especially in terms of generation quality. When $\lambda = 1$, only the timestep with the highest SSScore is selected. This setting also underperforms compared to selecting a set of timesteps with an appropriate threshold. The results suggest that choosing $\lambda$ in the range of [0.5, 0.8] achieves a favorable trade-off between erasure effectiveness and generation capability. In particular, $\lambda = 0.8$ shows the best overall performance. As a comparison, fine-tuning on 5 randomly selected timesteps leads to significantly higher FID scores, further demonstrating the advantage of the score-guided selection strategy.

**Impact of Core Training Components.** This section evaluates the impact of incorporating the quality regulation $\mathcal{L}_p$ and $\mathcal{L}_r$. Four experiments are conducted involving batch erasure of 20 celebrities, with results reported as the average CLIP and FID scores. The results are shown in Table 5. Compared to the baseline without regularization, the proposed method substantially enhances the post-erasure generation quality. Specifically, the FID score decreases from 8.05 to 6.65, represent-

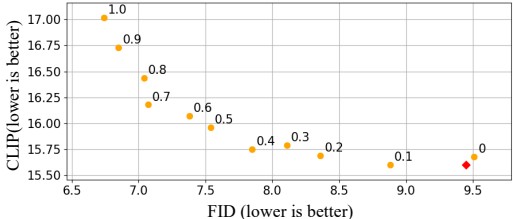

Figure 5: Impact of Step Selection threshold $\lambda$. The yellow dots and their corresponding numbers represent the experimental results under each $\lambda$ setting, while the red dot indicates the result of randomly selecting 5 timesteps.

Table 5: Impact of Regularization Loss on model's generative ability. The CLIP scores of the original model are included for reference.

| Setting | CLIP↓ | MSCOCO-30k FID↓ |
|---|---|---|
| Baseline (no regularization) | **10.27** | 8.05 |
| + $\mathcal{L}_p$ | 10.83 | 7.39 |
| + $\mathcal{L}_r$ | 10.47 | 7.06 |
| Ours(+ $\mathcal{L}_p$ + $\mathcal{L}_r$) | 10.91 | **6.65** |
| SD1.4 | 29.25 | - |

Table 6: Comparison of anchor selection strategies for concept erasure. Random anchors are randomly selected from {"sky", "ground", "forest", "street"}. Empty represents anchor-free operation. Ours employs heuristic selection rules for anchor assignment.

| Anchor Type | Airplane | | | | Automobile | | | | Bird | | | | Cat | | | | **Average** | | | |
|---|---|---|---|---|---|---|---|---|---|---|---|---|---|---|---|---|---|---|---|---|
| | $Acc_e$↓ | $Acc_s$↑ | $Acc_g$↓ | $H_o$↑ | $Acc_e$↓ | $Acc_s$↑ | $Acc_g$↓ | $H_o$↑ | $Acc_e$↓ | $Acc_s$↑ | $Acc_g$↓ | $H_o$↑ | $Acc_e$↓ | $Acc_s$↑ | $Acc_g$↓ | $H_o$↑ | $Acc_e$↓ | $Acc_s$↑ | $Acc_g$↓ | $H_o$↑ |
| SD v1.4 | 96.06 | 98.92 | 95.08 | – | 95.75 | 98.95 | 75.91 | – | 99.72 | 98.51 | 95.45 | – | 98.93 | 98.60 | 99.05 | – | 98.63 | 98.63 | 83.64 | – |
| random | 15.65 | 96.61 | 12.98 | 89.03 | 11.16 | 93.87 | 24.14 | 85.49 | 4.11 | 90.63 | 6.95 | **93.14** | 13.52 | 88.58 | 10.72 | 88.10 | 10.12 | 92.96 | 11.70 | 90.21 |
| empty | 15.79 | 90.52 | 12.76 | 87.25 | 14.71 | 98.36 | 18.95 | 87.64 | 11.4 | 93.6 | 20.11 | 86.99 | 7.37 | 97.97 | 17.96 | 90.38 | 14.09 | 95.84 | 15.93 | 88.17 |
| **Ours** | 4.37 | 95.32 | 2.66 | **96.09** | 2.77 | 98.80 | 11.54 | **94.60** | 9.72 | 97.82 | 18.49 | 89.37 | 0.49 | 98.56 | 4.63 | **97.78** | 2.71 | 98.00 | 8.71 | **95.33** |

ing a 17.4% improvement. Meanwhile, the CLIP score shows a modest increase from 10.27 to 10.91 (6.2%). Notably, the CLIP score remains well below that of the original model, indicating effective erasure. These results indicate that the introduction of quality regulation contributes notably to improving overall performance.

**Impact of Anchor choice.** For object erasure tasks where structural layout is established in early timesteps, we employ a **heuristic anchor selection** strategy. Specifically, we select anchor concepts that share high visual and structural similarity with the target (e.g., pairing "Cat" with "Dog", or "Automobile" with "Truck"), ensuring the erased trajectory transitions smoothly to a proximate natural image manifold. To systematically investigate the influence of anchor selection, we compare three strategies on CIFAR-10: (1) Ours (heuristic), (2) Random (selecting generic concepts like "Sky" or "Forest"), and (3) Empty (anchor-free).

Quantitative results in Table 6 show that heuristic method achieves the best overall performance, confirming that structurally aligned anchors minimize disturbance to the generative process. When using a sub-optimal anchor (random), our method still achieves performance superior to all baselines except MACE. Although slightly lower than MACE, it is important to note that MACE requires images along with SAM-generated masks as training data, whereas our method attains comparable effectiveness using only a single prompt. The Empty setting remains highly effective, significantly outperforming most baselines. This demonstrates that while our heuristic anchor enhances quality for structural concepts, the core robustness of SDErasure stems from the precise Step Selection and Quality Regulation. Further details are provided in Appendix I.2.

## 5 CONCLUSION

In this work, we investigate the denoising dynamics of diffusion models and demonstrate that concept erasure does not need to be applied uniformly across all timesteps. Instead, we show that selective erasure at specific timesteps can be both effective and efficient. Building on this insight, we propose SDErasure, a novel framework designed for adaptive and concept-specific erasure. First, Step Selection is proposed to pinpoint concept-specific key timesteps, while Score Rematching further improves alignment between the erased and anchor concepts. To preserve the model's generative capacity, we also introduce a quality regulation consisting of the early-preserve loss and the concept-retain loss. Experimental results demonstrate that the proposed method not only achieves effective concept erasure but also excels in two key aspects: enhancing related-concept quality and preserving unrelated-concept fidelity, thereby setting a new state-of-the-art in concept erasure.

ETHICS STATEMENT

All datasets used in this work are publicly available. Our study is dedicated to addressing the safety concerns of generative models. To demonstrate the effectiveness of different erasure methods, the paper unavoidably involves explicit content. However, we ensure that all such content has been properly obfuscated (e.g., blurred or masked) to prevent misuse and to minimize potential ethical risks.

REPRODUCIBILITY STATEMENT

We have made extensive efforts to ensure the reproducibility of our work. Detailed theoretical derivations and algorithmic pseudocode are provided in Appendix B and E. The hyperparameter configurations and training setups are documented in Section H. For the experimental studies, detailed procedures are provided in Section H.1 (object erasure), Section H.2 (celebrity erasure), Section H.3 (artistic style erasure), and Section H.4 (NSFW concept erasure). All code required to reproduce our results is included in the supplementary material.

ACKNOWLEDGMENTS

This work is supported by the National Natural Science Foundation of China (62425114, 62121002, U23B2028, 62472395), and the Fundamental and Interdisciplinary Disciplines Breakthrough Plan of the Ministry of Education of China (JYB2025XDXM103).

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

# A PRELIMINARIES

## A.1 DIFFUSION MODEL

Denoising Diffusion Probabilistic Models (DDPMs) establish a generative framework wherein a data sample $\mathbf{x}_0$ is progressively perturbed into standard Gaussian noise over $T$ discrete timesteps via a predefined forward diffusion process. The model is trained to learn a reverse denoising trajectory that reconstructs the original data by approximating the time-reversed dynamics of this process:

$$p_\theta(\mathbf{x}_{t-1} \mid \mathbf{x}_t) = \mathcal{N}\big(\mathbf{x}_{t-1}; \boldsymbol{\mu}_\theta(\mathbf{x}_t, t), \sigma_t^2 \mathbf{I}\big). \tag{16}$$

The network typically predicts the mean $\boldsymbol{\mu}_\theta$, often parameterized via a noise estimate $\epsilon_\theta$, and optionally the variance $\sigma_t^2$. In practice, $\sigma_t^2$ is usually fixed as $\beta_t$, and the model learns to predict the noise term:

$$\boldsymbol{\mu}_\theta(\mathbf{x}_t, t) = \frac{1}{\sqrt{\alpha_t}}\left(\mathbf{x}_t - \frac{\beta_t}{\sqrt{1 - \bar{\alpha}_t}} \epsilon_\theta(\mathbf{x}_t, t)\right). \tag{17}$$

Training is conducted by minimizing a simplified form of the variational bound:

$$\mathcal{L}_{\text{simple}} = \mathbb{E}_{\mathbf{x}_0, \epsilon, t}\Big[\|\epsilon - \epsilon_\theta(\mathbf{x}_t, t)\|^2\Big], \tag{18}$$

where $\mathbf{x}_t = \sqrt{\bar{\alpha}_t}\, \mathbf{x}_0 + \sqrt{1 - \bar{\alpha}_t}\, \epsilon$. This loss is derived from the Evidence Lower Bound (ELBO), a standard objective in variational inference. In the DDPM framework, the simplified loss $\mathcal{L}_{\text{simple}}$ corresponds to optimizing a reweighted version of the ELBO. The model learns to predict the noise $\epsilon$ added at each step of the forward process.

During sampling, the process begins with $\mathbf{x}_T \sim \mathcal{N}(\mathbf{0}, \mathbf{I})$ and recursively applies $p_\theta$ from $t = T$ to $1$ to generate $\mathbf{x}_0$. To improve conditional fidelity, classifier-free guidance combines conditional and unconditional score estimates:

$$\hat{\epsilon}_\theta = \epsilon_\theta(\mathbf{x}_t, t) + s\big(\epsilon_\theta(\mathbf{x}_t, t, \mathbf{c}) - \epsilon_\theta(\mathbf{x}_t, t)\big), \tag{19}$$

where $s > 1$ is the guidance scale and $\mathbf{c}$ denotes a conditioning embedding. This simple yet powerful mechanism allows a trade-off between sample diversity and adherence to the conditioning signal.

## A.2 SCORE FUNCTION

In score-based diffusion, the forward process is described by the SDE:

$$d\mathbf{x} = f(\mathbf{x}, t)dt + g(t)d\mathbf{w}, \tag{20}$$

with marginal density $p_t(\mathbf{x})$. The score function is defined as:

$$\mathbf{s}(\mathbf{x}, t) = \nabla\mathbf{x} \log p_t(\mathbf{x}), \tag{21}$$

which points toward regions of higher data density at each timestep. We approximate this with a neural network $\mathbf{s}_\theta(\mathbf{x}, t)$ trained via denoising score matching:

$$\mathcal{L}DSM(\theta) = \mathbb{E}_{t, \mathbf{x}_0, \mathbf{x}_t}\big[\lambda(t)|\mathbf{s}_\theta(\mathbf{x}_t, t) - \nabla\mathbf{x}_t \log p(\mathbf{x}_t|\mathbf{x}_0)|^2\big]. \tag{22}$$

In Diffusion Models, given a Gaussian distribution $\mathbf{x} \sim \mathcal{N}(\mu, \sigma^2\mathbf{I})$, the gradient of the log-probability density is represented as:

$$\nabla_{\mathbf{x}} \log p(\mathbf{x}) = \nabla_{\mathbf{x}}\left(-\frac{1}{2\sigma^2}(\mathbf{x} - \boldsymbol{\mu})^2\right) = -\frac{\mathbf{x} - \boldsymbol{\mu}}{\sigma^2} = -\frac{\epsilon}{\sigma}, \tag{23}$$

recall that $q(\mathbf{x}_t \mid \mathbf{x}_0) \sim \mathcal{N}(\sqrt{\bar{\alpha}_t}\mathbf{x}_0, (1 - \bar{\alpha}_t)\mathbf{I})$ and therefore,

$$\begin{aligned}
\mathbf{s}_\theta(\mathbf{x}_t, t) \approx \nabla_{\mathbf{x}_t} \log q(\mathbf{x}_t) &= \mathbb{E}_{q(\mathbf{x}_0)}\left[\nabla_{\mathbf{x}_t} \log q(\mathbf{x}_t \mid \mathbf{x}_0)\right] \\
&= \mathbb{E}_{q(\mathbf{x}_0)}\left[-\frac{\epsilon_\theta(\mathbf{x}_t, t)}{\sqrt{1 - \bar{\alpha}_t}}\right] \\
&= -\frac{\epsilon_\theta(\mathbf{x}_t, t)}{\sqrt{1 - \bar{\alpha}_t}}.
\end{aligned} \tag{24}$$

In summary, the noise predictor $\epsilon_\theta(\mathbf{x}_t, t)$ relates to the score via:

$$\mathbf{s}_\theta(\mathbf{x}_t, t) \approx -\frac{1}{\sigma_t}\epsilon_\theta(\mathbf{x}_t, t). \tag{25}$$

This correspondence enables Diffusion models to leverage the score function implicitly through $\epsilon_\theta$.

---

**Algorithm 1** Step Selection Based on Diffusion Classifier

---

**Input**: Image $x$, target concept $c_t$, anchor concept $c_a$
**Parameter**: Timestep set $T = \{1, 2, \ldots, 49\}$, iteration count $K$, threshold $\lambda$
**Output**: Relevant timestep set $\mathcal{T}$

 1: Initialize $\mathcal{T} \leftarrow \emptyset$
 2: **for** each $t \in T$ **do**
 3:     Initialize SSScores $\leftarrow \emptyset$
 4:     **for** $k = 1$ to $K$ **do**
 5:         $x_t \leftarrow \texttt{add\_noise}(x, \varepsilon, t)$
 6:         $\varepsilon_t \leftarrow \varepsilon_\theta(x_t, t, c_t)$
 7:         $\varepsilon_a \leftarrow \varepsilon_\theta(x_t, t, c_a)$
 8:         $\mathcal{S}_t \leftarrow \frac{\exp(-(\varepsilon_t - \varepsilon)^2)}{\exp(-(\varepsilon_t - \varepsilon)^2) + \exp(-(\varepsilon_a - \varepsilon)^2)}$
 9:         Append $\mathcal{S}_t$ to SSScores
10:     **end for**
11:     Compute $\bar{\mathcal{S}}_t \leftarrow \mathrm{mean}(\text{SSScores})$
12: **end for**
13: Normalize $\{\bar{\mathcal{S}}_t\}$ to $[0, 1]$
14: **for** each $t \in T$ **do**
15:     **if** $\bar{\mathcal{S}}_t > \lambda$ **then**
16:         Add $t$ to $\mathcal{T}$
17:     **end if**
18: **end for**
19: **return** $\mathcal{T}$

---

## B    PSEUDOCODE OF STEP SELECTION

The pseudocode for the Step Selection algorithm is presented in Algorithm 1. At each timestep, the model predicts noise conditioned on both the target concept (to be erased) and the anchor concept (a guiding replacement concept that redirects generation). A diffusion classifier is employed to compute the classification probability for each prediction, based on which the Step Separability Score (SSScore) is computed and normalized across timesteps. Timesteps with higher scores correspond to greater divergence between the denoising trajectories of the target and anchor concepts.

## C    LIMITATIONS

For object erasure tasks, SDErasure incorporates an anchor concept during training. As with previous anchor-based approaches, performance may vary slightly with different anchor choices. Additional heuristics and ablations, including human-selected, random, and empty anchors, are presented in Section I.

## D    TIME COST OF SSSCORE

The Step Selection (SSScore calculation) is a one-time pre-processing step, not a recurring cost during the main training loop or at inference time.

The overhead of this step is minimal. As detailed in Section H.1, for object erasure on CIFAR-10, the entire erasure process takes only about 20 minutes on a single RTX 4090, with the **Step Selection algorithm consuming only less than 3 minutes** of that time. By pinpointing the few critical timesteps, we avoid fine-tuning across all timesteps, which makes our overall training process approximately **75% faster** than previous trajectory-aware methods like ANT. Thus, the slight pre-computation overhead is far outweighed by the significant gain in training efficiency.

### ACCELERATION STRATEGIES

Furthermore, this pre-computation step can be significantly accelerated, particularly for large-scale applications. We propose two effective strategies.

Table 7: Comparison of Generation Results for Different Artist Styles using Sparse Sampling for SSScore calculation. 'original' refers to sampling at every timestep. 'sparse sampling (k)' refers to sampling every $k$ steps.

| Concept | Method | $CLIP_{target} \downarrow$ | $FID_{coco} \downarrow$ | $CLIP_{coco} \uparrow$ | Time cost |
|---|---|---|---|---|---|
| Van Gogh | original | 21.17 | 7.02 | 31.17 | $161.94s$ |
| | sparse sampling (3) | 20.43 | 7.62 | 31.12 | $57.17s$ |
| | sparse sampling (5) | 18.85 | 8.44 | 31.13 | $34.25s$ |
| Picasso | original | 20.98 | 9.06 | 31.13 | $164.62s$ |
| | sparse sampling (3) | 16.77 | 9.98 | 31.08 | $57.55s$ |
| | sparse sampling (5) | 18.02 | 9.25 | 31.06 | $33.88s$ |
| Monet | original | 20.78 | 8.09 | 31.17 | $164.02s$ |
| | sparse sampling (3) | 18.60 | 8.40 | 31.11 | $57.37s$ |
| | sparse sampling (5) | 20.22 | 7.87 | 31.12 | $33.79s$ |
| Average | original | 20.98 | **8.06** | **31.16** | $163.53s$ |
| | sparse sampling (3) | **18.60** | 8.67 | 31.10 | $57.36s$ |
| | sparse sampling (5) | 19.03 | 8.52 | 31.10 | **33.97s** |

SPARSE SAMPLING

As shown in Figure 6, the SSScore distribution across timesteps is relatively smooth and continuous. This suggests that calculating the score at every single timestep is unnecessary. We can employ **sparse sampling** (e.g., computing scores every 3 or 5 steps) and use interpolation to estimate the full curve.

To validate this, we conducted additional experiments to explore the influence of sparse sampling on model performance for artist style erasure. The experimental results are shown in Table 7. $CLIP_{target}$ represents the similarity between the target concept and its corresponding prompt, indicating the effectiveness of the erasure. $FID_{coco}$ and $CLIP_{coco}$ measure the generative performance of the erasure model on the MSCOCO dataset.

As shown in Table 7, sparse sampling performs comparably to the original strategy (sampling every step). On average, it produces slightly stronger erasure (lower target CLIP) and marginally weaker general-purpose generation (slightly higher FID). Importantly, sparse sampling at 5-step intervals **reduces the pre-computation time cost by ~80%** (from 163.5s to 34s), demonstrating clear practical value for accelerating the process.

CONCEPT CLUSTERING FOR MULTI-CONCEPT ERASURE

For large-scale multi-concept erasure, a second acceleration strategy can be employed. Our key observation is that structural concepts primarily emerge in the early–mid denoising stages, whereas fine-grained concepts (like faces or styles) appear in the mid–late stages.

Based on this, we can first use a Large Language Model (LLM) to cluster the concepts to be removed (e.g., into "structural" vs. "fine-grained" groups). Concepts within the same cluster can then **share a single SSScore calculation**. This dramatically reduces the pre-computation cost from $O(N_{concepts})$ to $O(N_{clusters})$. In fact, the multi-concept erasure results in Table 12 of the appendix were trained using such a shared set of SSScore and optimal steps, showing that this approach achieves both high efficiency and accuracy.

# E  DERIVATION OF SCORE REMATCHING LOSS

In image generation, the score function serves as a vector field that guides samples toward regions of high data density, thereby acting as the fundamental mechanism underlying the generative process.

The primary objective of the proposed method is to align the predicted score function of the fine-tuned model, conditioned on the target concept, with the original model's score function under an anchor concept:

$$s_\theta(x_t, c_t, t) \leftarrow s_{\theta^*}(x_t, c_a, t), \tag{26}$$

Table 8: The synonyms and anchor concepts for the ten object classes in the CIFAR-10 dataset.

| Object Classes | Airplane | Automobile | Bird | Cat | Deer | Dog | Frog | Horse | Ship | Truck |
|---|---|---|---|---|---|---|---|---|---|---|
| **Synonyms** | Aircraft Plane Jet | Car Vehicle Motorcar | Avian Fowl Winged Creature | Feline Kitty Housecat | Hart Stag Doe | Canine Pooch Hound | Amphibian Anuran Tadpole | Equine Steed Mount | Vessel Boat Watercraft | Lorry Rig Hauler |
| **Anchors** | Automobile | Truck | Cat | Dog | Horse | Cat | Cat | Deer | Airplane | Ship |

where $\theta$ denotes the parameters of the fine-tuned model, $\theta^*$ denotes the parameters of the original model, $c_t$ denotes the target concept, and $c_a$ denotes the anchor concept.

Furthermore, the fine-tuned model is encouraged to deviate from the original model's score function under the target concept. To achieve this, a negative guidance term is introduced into Equation (26):

$$s_\theta(x_t, c_t, t) \leftarrow s_{\theta^*}(x_t, c_a, t) \; - \; \eta \, \sigma_s(x_t, c_t, c_a, t), \tag{27}$$

where the guidance is defined as:

$$\sigma_s(x_t, c_t, c_a, t) = s_{\theta^*}(x_t, c_t, t) \; - \; s_{\theta^*}(x_t, c_a, t). \tag{28}$$

Applying the relation between $\epsilon_\theta$ and $s_\theta$ in Equation (25), we obtain the following training objective for Stable Diffusion:

$$\epsilon_\theta(x_t, c_t, t) \leftarrow \epsilon_{\theta^*}(x_t, c_a, t) \; - \; \eta \, \sigma(x_t, c_t, c_a, t), \tag{29}$$

with

$$\sigma(x_t, c_t, c_a, t) = \epsilon_{\theta^*}(x_t, c_t, t) \; - \; \epsilon_{\theta^*}(x_t, c_a, t). \tag{30}$$

## F  SSSCORE ANALYSIS AND VALIDATION

To demonstrate the effectiveness of SSScore and its relationship with concept variation, we compute the SSScore for Van Gogh Style and Picasso Style across different timesteps and visualize the results as bar charts in Figure 6. Van Gogh Style typically features rich brushstrokes and rapid color transitions, with a predominance of high-frequency detail information. Correspondingly, SSScore identifies the critical timesteps for Van Gogh Style generation in the middle-to-late stages of denoising, which precisely aligns with the period when high-frequency details are synthesized. In contrast, Picasso Style is characterized by large color blocks and a prevalence of low-frequency structural information, resulting in higher SSScore values during the early-to-middle stages of denoising. This experiment validates that SSScore can adaptively select critical timesteps for concept generation based on the distinctive characteristics of different concepts, thereby demonstrating the effectiveness of our Step Selection mechanism.

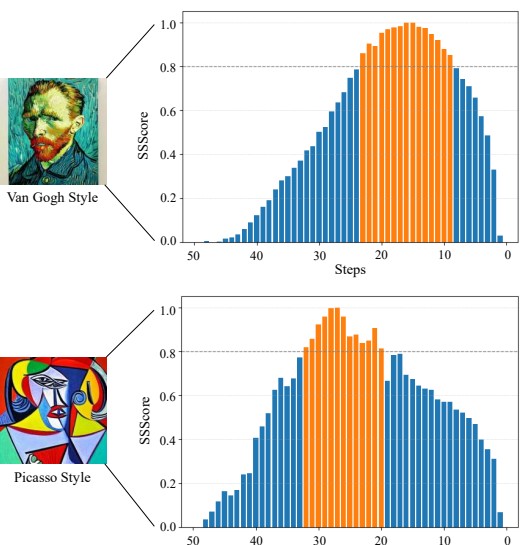

Figure 6: SSScores for Van Gogh Style and Picasso Style, with the anchor concept set to empty in both cases.

In Figure 7, we present bar charts of SSScores for a broader range of concepts, including objects, celebrities, artistic styles, and nudity. Consistent with our earlier observations, structural concepts such as objects and nudity exhibit high SSScores primarily in the early to mid denoising stages, whereas fine-grained concepts such as human faces and artistic styles tend to peak in the mid to late stages. These results further substantiate the effectiveness of SSScore.

To verify that the identified timesteps are indeed optimal, we conduct an ablation study by evenly dividing all denoising steps into ten groups. Group 1 corresponds to steps [45, 49] and Group 10 to steps [1, 5]. We erase the Picasso Style using each group of timesteps and evaluate both the erasure performance (CLIP) and the generation quality (FID). The results in Figure 8 show that Group 5 (steps [25, 30]) achieves the best erasure performance, which coincides with the peak of the SSScore, confirming that the timesteps selected by SSScore are optimal.

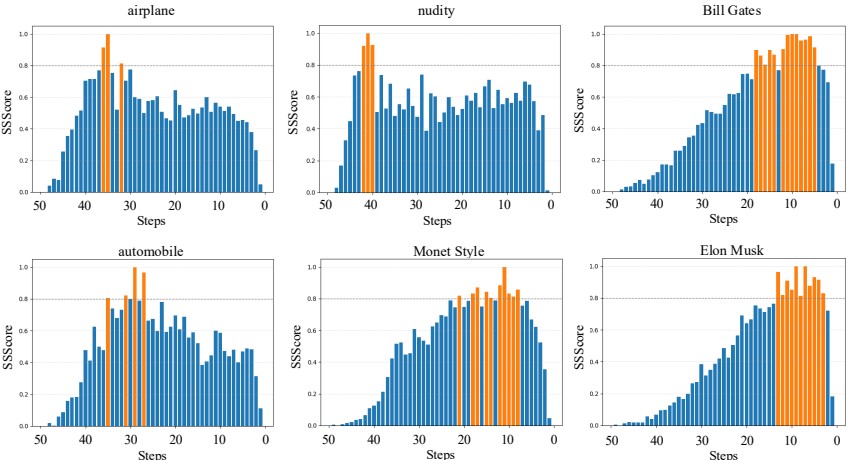

Figure 7: SSScores for different categories of concepts.

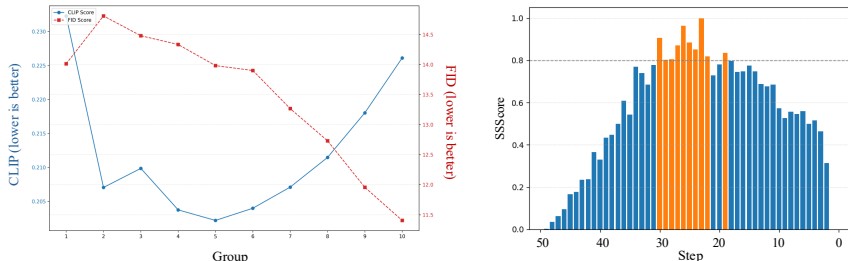

Figure 8: Erasure performance of the Picasso Style across different timestep groups, together with the corresponding SSScore bar charts. The best performance occurs in Group 5 (timesteps 25–30), which aligns with the SSScore peak.

## G  GENERALIZABILITY OF STEP SELECTION

In this section, we study whether the Step Selection procedure can still identify the correct critical timesteps when the total number of diffusion timesteps changes. Using Van Gogh and Picasso as examples, we visualize SSScore bar charts for total timesteps of $T = 50$ (default), $T = 100$, and $T = 200$; the results are shown in Figure 9.

The plots indicate that the SSScore profile remains consistent in shape when $T$ changes. This observation suggests that SSScore identifies the *relative generative phase* (i.e., the effective noise level) rather than an absolute timestep index. Consequently, the Step Selection strategy exhibits strong generalizability across different discretizations of the diffusion timeline.

## H  IMPLEMENTATION DETAILS

**Hyperparameters**. In practice, K is set to 20, $\eta$ is set to 1, $\beta_1$ and $\beta_2$ are set to 0.1 to ensure that the model continues to perform well on the main task. The selection of fine-tuned parameters is adapted to the characteristics of different erasure scenarios. Specifically, $W_q$ in the cross-attention

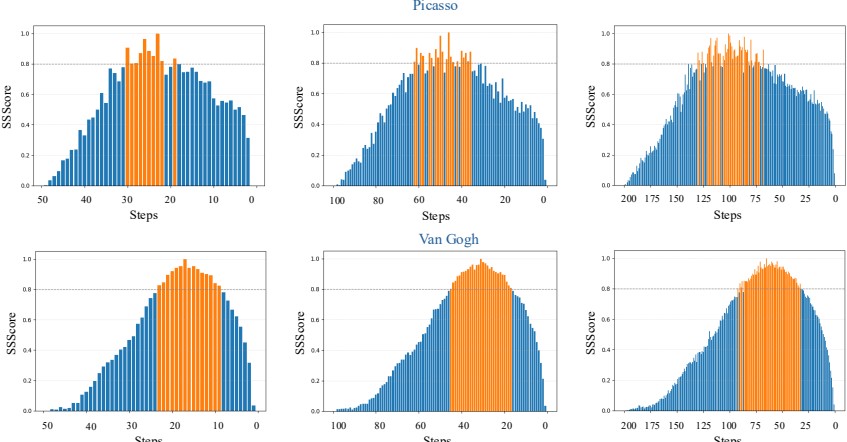

Figure 9: SSScores when the total number of timesteps changes. The left column corresponds to a total time step of 50, the middle column corresponds to a total time step of 100, and the right column corresponds to a total time step of 200.

modules is tuned for removing concepts such as celebrities and artistic styles, while the entire cross-attention modules are updated for object-level erasure. For global concepts, such as explicit content, all attention layers are selected to achieve broader semantic removal.

## H.1 OBJECT ERASURE DETAILS

To evaluate the generalization capability of concept erasure, this work adopts the MACE protocol by selecting three synonyms for each of the ten CIFAR-10 classes, as detailed in Table 8. For each class, a separate SDErasure model is trained using diverse prompts generated via the DeepSeek-v3 API to enhance robustness against prompt variation. A Step Selection threshold of $\lambda = 0.5$ is employed to ensure stability across different anchor concepts. Fine-tuning is conducted on Stable Diffusion v1.4 for 200 steps with a learning rate of 2e-5. The entire training process is memory-efficient, requiring only 14GB of GPU memory on a single RTX 4090, and completes within approximately 20 minutes, including around 3 minutes for Step Selection. Compared to previous trajectory-aware method ANT, our approach reduces training time by approximately 75%.

For each target concept, a distinct CIFAR-10 class is designated as the anchor to evaluate the effectiveness of concept alteration. The full experimental configuration is summarized in Table 8. To compare different erasure methods, the fine-tuned models are evaluated along three key dimensions:

- **Efficacy:** For the erased target concept, 200 images are generated using the prompt "a photo of the {*erased class*}."
- **Generality:** For the three synonym concepts of the erased target, 200 images are generated each using the prompt "a photo of the {*synonymous class*}," for a total of 600 images.
- **Specificity:** For the nine remaining concepts, 200 images are generated each using the prompt "a photo of the {*unaltered class*}," for a total of 1,800 images.

## H.2 CELEBRITY ERASURE DETAILS

The Step Selection threshold is set to $\lambda = 0.8$. Fine-tuning is performed in an anchor-free setting, where the anchor is omitted. The training is conducted for 300 steps with a learning rate of 1e-4. To evaluate the erasure effectiveness for each celebrity identity, the performance is assessed along three dimensions:

- **Efficacy:** 100 images are generated with the prompt "a photo of {*erased celebrity*}."
- **Specificity:** 100 images are generated with prompts "a photo of {*neighbor celebrity*}" for other celebrities and compute the LPIPS score against the original model's outputs. Moreover, FID and CLIP scores are evaluated on a 3k subset of MSCOCO-30k.

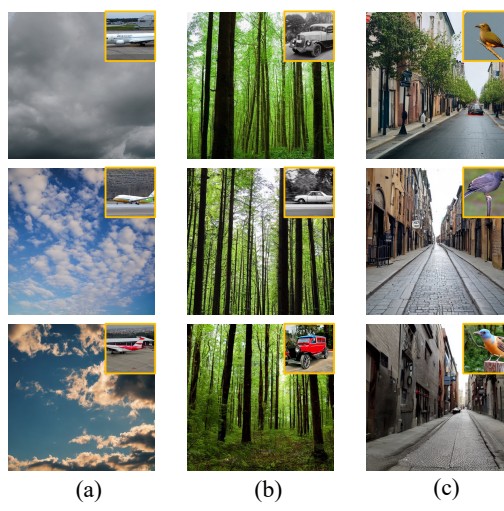

Figure 10: Visualization of Erasure with Generic Anchors. (a) Erasing "airplane" to "sky". (b) Erasing "automobile" to "forest". (c) Erasing "bird" to "street".

Table 9: CLIP and FID scores for erasing individual celebrity identities using SDErasure. Each row reports the results for a specific celebrity. The bottom two rows summarize the average performance across all identities and the corresponding CLIP score of the original SD1.4 model for comparison.

| Name | CLIP↓ | FID↓ | Name | CLIP↓ | FID↓ |
|------|-------|------|------|-------|------|
| Anne Hathaway | 10.79 | 6.59 | Andy Samberg | 10.35 | 6.40 |
| Adam Driver | 7.96 | 6.66 | Alec Baldwin | 9.49 | 6.71 |
| Amy Poehler | 10.90 | 6.56 | Avril Lavigne | 12.07 | 6.64 |
| Aaron Paul | 4.00 | 6.53 | Arnold Schwarzenegger | 13.15 | 7.14 |
| Amanda Seyfried | 11.53 | 6.52 | Andrew Garfield | 14.51 | 6.41 |
| Amy Schumer | 13.85 | 6.48 | Amy Winehouse | 12.36 | 6.72 |
| Anna Faris | 13.46 | 6.96 | Amber Heard | 12.28 | 6.19 |
| Adriana Lima | 9.26 | 7.65 | Aretha Franklin | 11.39 | 6.65 |
| Amy Adams | 9.03 | 6.65 | Anjelica Huston | 7.56 | 6.48 |
| Angelina Jolie | 16.72 | 6.67 | Anna Kendrick | 7.52 | 6.43 |
| **Average** | | | CLIP: 10.91 | FID: 6.65 | |
| **SD1.4** | | | CLIP: 28.56 | FID: – | |

## H.3 ARTIST STYLE ERASURE DETAILS

All training hyperparameters are the same as in Celebrity Erasure. The erasure performance is assessed along two dimensions:

- **Efficacy:** 100 images are generated with the prompt "a painting in the style of {*erased style*}."
- **Specificity:** FID and CLIP scores are evaluated on a 3k subset of MSCOCO-30k.

## H.4 EXPLICIT CONCEPT ERASURE DETAILS

The Step Selection threshold is set to $\lambda = 0.8$. The anchor is set to "a person wearing clothes". The training runs for 200 steps with a learning rate of 2e-5. The sensitive-content erasure performance is assessed along two dimensions:

- **Efficacy:** NudeNet is applied to count sensitive outputs on the i2p dataset.
- **Specificity:** FID and CLIP scores are evaluated on a 3k subset of MSCOCO-30k.

## I CHOICES OF ANCHOR CONCEPT

Overall, for the erasure of celebrities and artistic styles, we recommend employing an anchor-free training approach by setting the anchor concept to empty. This is because for such fine-grained semantic details, SSScore tends to select timesteps in the middle-to-late stages of the diffusion process. During these stages, fine-tuning has minimal impact on the overall image structure, allowing effective erasure to be achieved without the need for anchoring. In contrast, for structural concepts such as objects, SSScore typically identifies timesteps in the early-to-middle stages, where employing an anchor helps ensure that the generated images remain within the natural image manifold. We provide detailed discussions on the anchor selection rationale in the following Sections I.1 and I.2.

## I.1 CELEBRITY AND STYLE ERASURE

By default, setting the anchor concept to an empty prompt is sufficient for effective concept erasure, while preserving semantically unrelated attributes such as structure and pose. However, when finer control over the generation process is desired, a specific anchor concept can be provided to steer the

Table 10: Comparison of anchor selection strategies for concept erasure. Results of the remaining six classes are reported. Empty represents anchor-free operation. Ours employs heuristic selection rules for anchor assignment.

| Anchor Type | Deer | | | | Dog | | | | Frog | | | | Horse | | | | Ship | | | | Truck | | | |
|---|---|---|---|---|---|---|---|---|---|---|---|---|---|---|---|---|---|---|---|---|---|---|---|---|
| | $Acc_e\downarrow$ | $Acc_s\uparrow$ | $Acc_g\downarrow$ | $H_o\uparrow$ | $Acc_e\downarrow$ | $Acc_s\uparrow$ | $Acc_g\downarrow$ | $H_o\uparrow$ | $Acc_e\downarrow$ | $Acc_s\uparrow$ | $Acc_g\downarrow$ | $H_o\uparrow$ | $Acc_e\downarrow$ | $Acc_s\uparrow$ | $Acc_g\downarrow$ | $H_o\uparrow$ | $Acc_e\downarrow$ | $Acc_s\uparrow$ | $Acc_g\downarrow$ | $H_o\uparrow$ | $Acc_e\downarrow$ | $Acc_s\uparrow$ | $Acc_g\downarrow$ | $H_o\uparrow$ |
| SD v1.4 | 99.87 | 98.49 | 70.02 | – | 98.74 | 98.62 | 98.25 | – | 99.93 | 98.49 | 92.04 | – | 99.78 | 98.50 | 45.74 | – | 98.64 | 98.63 | 63.16 | – | 98.89 | 98.60 | 95.00 | – |
| random | 18.49 | 92.8 | 11.19 | 87.45 | 9.36 | 91.01 | 11.66 | 89.98 | 11.52 | 96.67 | 15.18 | 89.72 | 7.99 | 88.09 | 9.58 | 90.14 | 6.75 | 95.2 | 12.19 | 91.98 | 2.64 | 96.18 | 2.39 | **97.05** |
| empty | 13.31 | 97.40 | 8.68 | 91.60 | 8.59 | 98.49 | 14.01 | 91.68 | 24.91 | 95.36 | 11.21 | 85.12 | 12.68 | 95.66 | 12.14 | 90.12 | 12.38 | 94.12 | 22.16 | 86.00 | 19.71 | 96.90 | 21.36 | 84.53 |
| **Ours** | 2.27 | 98.29 | 4.17 | **97.27** | 1.03 | 98.17 | 1.26 | **98.63** | 3.04 | 97.73 | 19.34 | **91.06** | 0.63 | 98.30 | 12.45 | **94.76** | 0.37 | 98.56 | 7.30 | **96.87** | 2.44 | 98.52 | 5.26 | 96.91 |

Table 11: **Evaluation of Category Erasure on CIFAR-10.** Results for the remaining six categories are reported. $Acc_e$ represents the accuracy for the erased class. $Acc_s$ represents the accuracy for unaltered classes. $Acc_g$ represents the accuracy for synonyms of the erased class. Higher $H_0$ indicates better overall concept erasure performance. All values are expressed as percentages (%). The original model's performance is included for reference.

| Method | Deer | | | | Dog | | | | Frog | | | | Horse | | | | Ship | | | | Truck | | | |
|---|---|---|---|---|---|---|---|---|---|---|---|---|---|---|---|---|---|---|---|---|---|---|---|---|
| | $Acc_e\downarrow$ | $Acc_s\uparrow$ | $Acc_g\downarrow$ | $H_o\uparrow$ | $Acc_e\downarrow$ | $Acc_s\uparrow$ | $Acc_g\downarrow$ | $H_o\uparrow$ | $Acc_e\downarrow$ | $Acc_s\uparrow$ | $Acc_g\downarrow$ | $H_o\uparrow$ | $Acc_e\downarrow$ | $Acc_s\uparrow$ | $Acc_g\downarrow$ | $H_o\uparrow$ | $Acc_e\downarrow$ | $Acc_s\uparrow$ | $Acc_g\downarrow$ | $H_o\uparrow$ | $Acc_e\downarrow$ | $Acc_s\uparrow$ | $Acc_g\downarrow$ | $H_o\uparrow$ |
| FMN | 98.95 | 94.13 | 60.24 | 3.04 | 97.64 | 98.12 | 96.95 | 3.94 | 91.60 | 94.59 | 63.61 | 19.10 | 99.63 | 93.14 | 46.61 | 1.10 | 97.97 | 98.21 | 96.75 | 3.70 | 97.64 | 97.86 | 95.37 | 4.62 |
| UCE | 11.88 | 98.39 | 8.94 | 92.34 | 13.22 | 98.69 | 14.63 | 89.90 | 20.86 | 98.32 | 18.50 | 85.53 | 4.66 | 98.32 | 12.70 | 93.42 | 6.13 | 98.41 | 21.44 | 89.44 | 20.58 | 98.16 | 50.00 | 70.13 |
| ESD-u | 18.14 | 73.81 | 6.93 | 82.17 | 27.03 | 89.75 | 28.52 | 77.24 | 12.32 | 88.05 | 7.62 | 89.32 | 17.69 | 82.23 | 9.89 | 84.73 | 18.38 | 94.32 | 15.93 | 86.33 | 26.11 | 85.35 | 21.47 | 78.98 |
| MACE | 13.47 | 97.71 | 6.08 | 92.48 | 11.07 | 96.77 | 10.86 | 91.47 | 11.45 | 97.75 | 13.08 | 90.83 | 4.89 | 97.48 | 7.85 | **94.86** | 8.58 | 98.56 | 14.40 | 91.56 | 7.29 | 98.38 | 9.38 | 93.79 |
| RECE | 9.62 | 98.27 | 7.51 | 93.60 | 17.72 | 98.51 | 26.78 | 83.55 | 11.69 | 98.43 | 10.23 | **91.96** | 9.98 | 98.10 | 14.61 | 90.87 | 21.92 | 98.41 | 25.61 | 82.39 | 30.42 | 98.43 | 40.69 | 72.48 |
| ANT | 3.61 | 98.46 | 2.5 | **97.44** | 17.22 | 98.56 | 20.12 | 86.34 | 23.99 | 98.42 | 10.8 | 86.89 | 12.93 | 98.47 | 11.65 | 91.02 | 13.74 | 98.55 | 21.39 | 87.05 | 22.04 | 98.47 | 21.93 | 83.82 |
| **Ours** | 2.27 | 98.29 | 4.17 | 97.27 | 1.03 | 98.17 | 1.26 | **98.63** | 3.04 | 97.73 | 19.34 | 91.06 | 0.63 | 98.30 | 12.45 | 94.76 | 0.37 | 98.56 | 7.30 | **96.87** | 2.44 | 98.52 | 5.26 | **96.91** |
| SD v1.4 | 99.87 | 98.49 | 70.02 | – | 98.74 | 98.62 | 98.25 | – | 99.93 | 98.49 | 92.04 | – | 99.78 | 98.50 | 45.74 | – | 98.64 | 98.63 | 63.16 | – | 98.89 | 98.60 | 95.00 | - |

resulting image. For example, when erasing the concept "Taylor Swift", using an anchor such as "a young woman" helps guide the model toward more targeted and semantically aligned replacements.

## I.2 OBJECT ERASURE

For concept erasure involving specific object categories, we recommend selecting anchor concepts that share similar visual characteristics with the target. For example, "cat" and "dog", or "deer" and "horse" exhibit comparable appearances and thus serve as effective anchor pairs. In the case of structural object categories, such as "automobile", an anchor concept like "truck" can provide a structurally coherent reference. However, some concepts may lack suitable anchors with the afore-mentioned characteristics. In such cases, generic concepts such as "sky", "forest" or "street" can be used as anchors to provide broad semantic guidance as shown in Figure 10.

## I.3 QUANTITATIVE RESULTS ON ANCHOR SELECTION

Table 10 presents quantitative results comparing random selection of generic concepts as anchors (random), anchor-free operation (empty), and heuristic anchor selection (ours). Random anchor selection achieves effective erasure performance, but the semantic distance between generic and target concepts adversely affects generation quality, as evidenced by a lower $Acc_s$ of 92.96 compared to 95.84 and 98.00 for the other two configurations. Setting the anchor to empty also demonstrates satisfactory erasure capabilities, though inferior to heuristic selection. In summary, our proposed selection strategy achieves optimal erasure performance, while both anchor-free and random anchor approaches demonstrate competitive advantages over existing methods.

Table 12: CLIP similarity scores for 20 concepts after multi-concept erasure using a single SDErasure model. The bottom two rows report the average performance and the baseline CLIP score of the original SD1.4 model.

| Concept | CLIP↓ | Concept | CLIP↓ |
|---|---|---|---|
| Aaron Paul | 19.28 | Andrew Garfield | 18.63 |
| Adam Driver | 17.01 | Andy Samberg | 20.01 |
| Adriana Lima | 17.79 | Angelina Jolie | 17.28 |
| Alec Baldwin | 15.49 | Anjelica Huston | 17.11 |
| Amanda Seyfried | 15.18 | Anna Faris | 21.16 |
| Amber Heard | 14.88 | Anna Kendrick | 16.25 |
| Amy Adams | 15.77 | Anne Hathaway | 15.91 |
| Amy Poehler | 17.61 | Aretha Franklin | 18.28 |
| Amy Schumer | 16.14 | Arnold Schwarzenegger | 19.32 |
| Amy Winehouse | 19.11 | Avril Lavigne | 17.05 |
| Average | CLIP: 17.46 | FID: 8.82 | |
| SD1.4 | CLIP: 28.56 | FID: - | |

Table 13: Detailed correspondence between erasure categories, concepts, and their visualizations.

| Category | Concept | Figure Index |
|---|---|---|
| Nudity Erasure | Nudity | Figure 11 |
| Celebrity Erasure | Elon Musk | Figure 12 |
| | Donald Trump | Figure 12 |
| | Taylor Swift | Figure 12 |
| Object Erasure | Airplane | Figure 13 |
| | Automobile | Figure 14 |
| | Bird | Figure 15 |
| | Cat | Figure 16 |
| | Deer | Figure 17 |
| | Dog | Figure 18 |
| | Frog | Figure 19 |
| | Horse | Figure 20 |
| | Ship | Figure 21 |
| | Truck | Figure 22 |

Table 14: Comparison of concept erasure methods on multi-concept erasure task. The CLIP score of the original model is provided as a reference.

| Method | Average CLIP↓ | FID↓ |
|---|---|---|
| UCE | 23.24 | 16.87 |
| MACE | 21.35 | 14.90 |
| ANT | 21.74 | 14.18 |
| **Ours** | **17.46** | **8.82** |
| Original | 28.56 | – |

## J    ADDITIONAL EXPERIMENTS

### J.1    ADDITIONAL CELEBRITY ERASURE

To evaluate the generalizability of the proposed method, additional experiments are conducted on a broader set of celebrity identities. As shown in Table 9, CLIP scores consistently decrease after erasure, indicating effective concept removal, while FID scores remain low (around 6.5), suggesting preserved image quality. These results demonstrate that the method scales well to diverse concepts.

### J.2    ERASING THE CIFAR-10 CLASSES

Table 11 presents the complete results of erasing the remaining six object categories in the CIFAR-10 dataset. SDErasure consistently achieves the highest harmonic mean, attributed to its concept-specific training strategy that adaptively shifts the denoising trajectories toward anchor concepts at critical timesteps.

### J.3    MULTI-CONCEPT ERASURE

To assess the model's capacity for multi-concept erasure, 20 concepts are removed simultaneously within a single model. Table 12 presents the CLIP scores for each concept and the overall FID after erasure. Table 14 presents the performance of different methods on multi-concept erasure tasks under the same experimental setup. While CLIP similarities increase compared to single-concept settings, they remain notably lower than those of the original model. Additionally, although erasing 20 concepts simultaneously, the FID score only modestly increases (from 6.65 to 8.82), and still outperforms previous methods as shown in Table 14. These results highlight the effectiveness of SDErasure in multi-concept erasure scenarios. This advantage stems from the design of SDErasure,

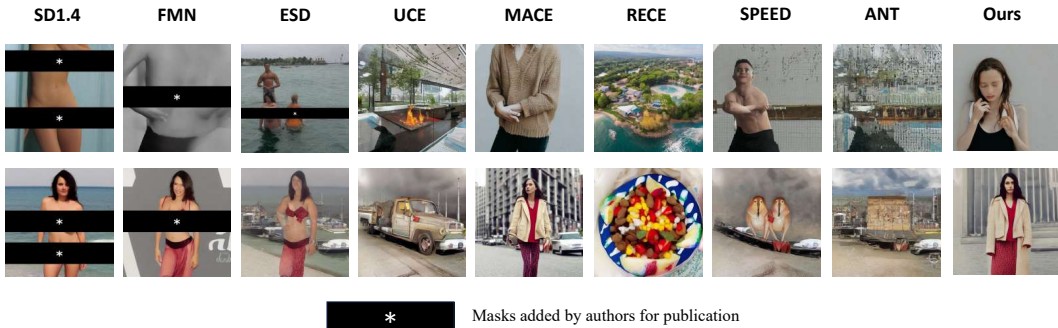

| SD1.4 | FMN | ESD | UCE | MACE | RECE | SPEED | ANT | Ours |

|   *   | Masks added by authors for publication |

Figure 11: Qualitative results on nudity erasure. The images on the same row are generated using the same random seed.

which focuses fine-tuning on low-noise timesteps when erasing celebrity concepts. Such a strategy mitigates interference among multiple concepts during simultaneous erasure, thereby maintaining erasure effectiveness while substantially improving the model's generative performance.

### J.4 ADDITIONAL QUALITATIVE RESULTS

In this section, we provide comprehensive qualitative results across a variety of concept erasure scenarios. Table 13 summarizes the mapping between each experiment and its corresponding visualization results.

## K USE OF LARGE LANGUAGE MODELS

The authors acknowledge the use of large language models (LLMs) in the preparation of this manuscript. Specifically, LLMs were employed for language editing and writing refinement purposes, including but not limited to grammar correction, sentence restructuring, and stylistic improvements to enhance the clarity and readability of the text. All technical content, experimental design, data analysis, results interpretation, and scientific conclusions presented in this work are entirely the authors' own contributions. The use of LLMs was limited to linguistic assistance and did not involve the generation of research ideas, methodological approaches, or substantive content. All factual claims and technical details have been independently verified by the authors.

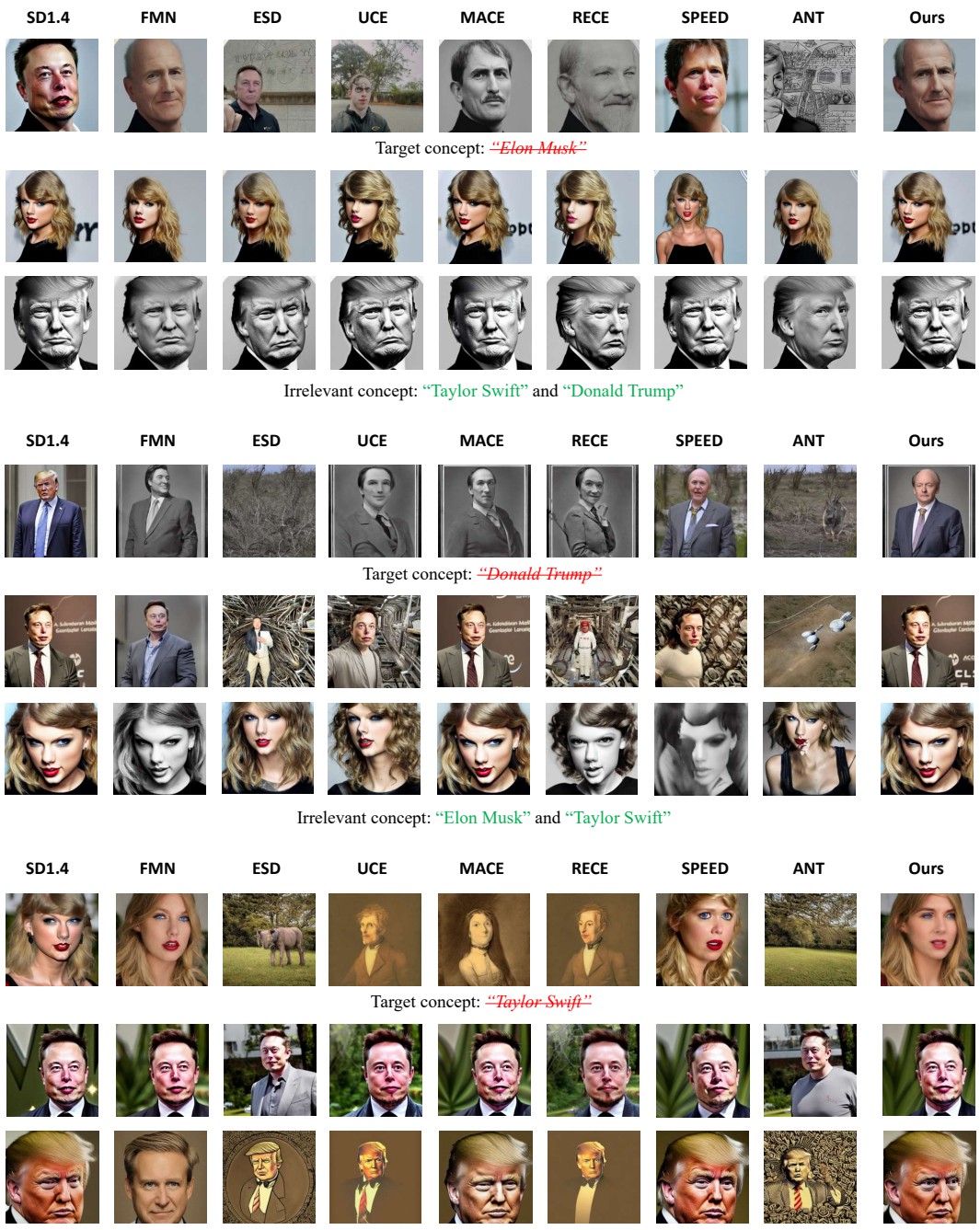

Figure 12: Qualitative results on celebrity erasure. The images on the same row are generated using the same random seed.

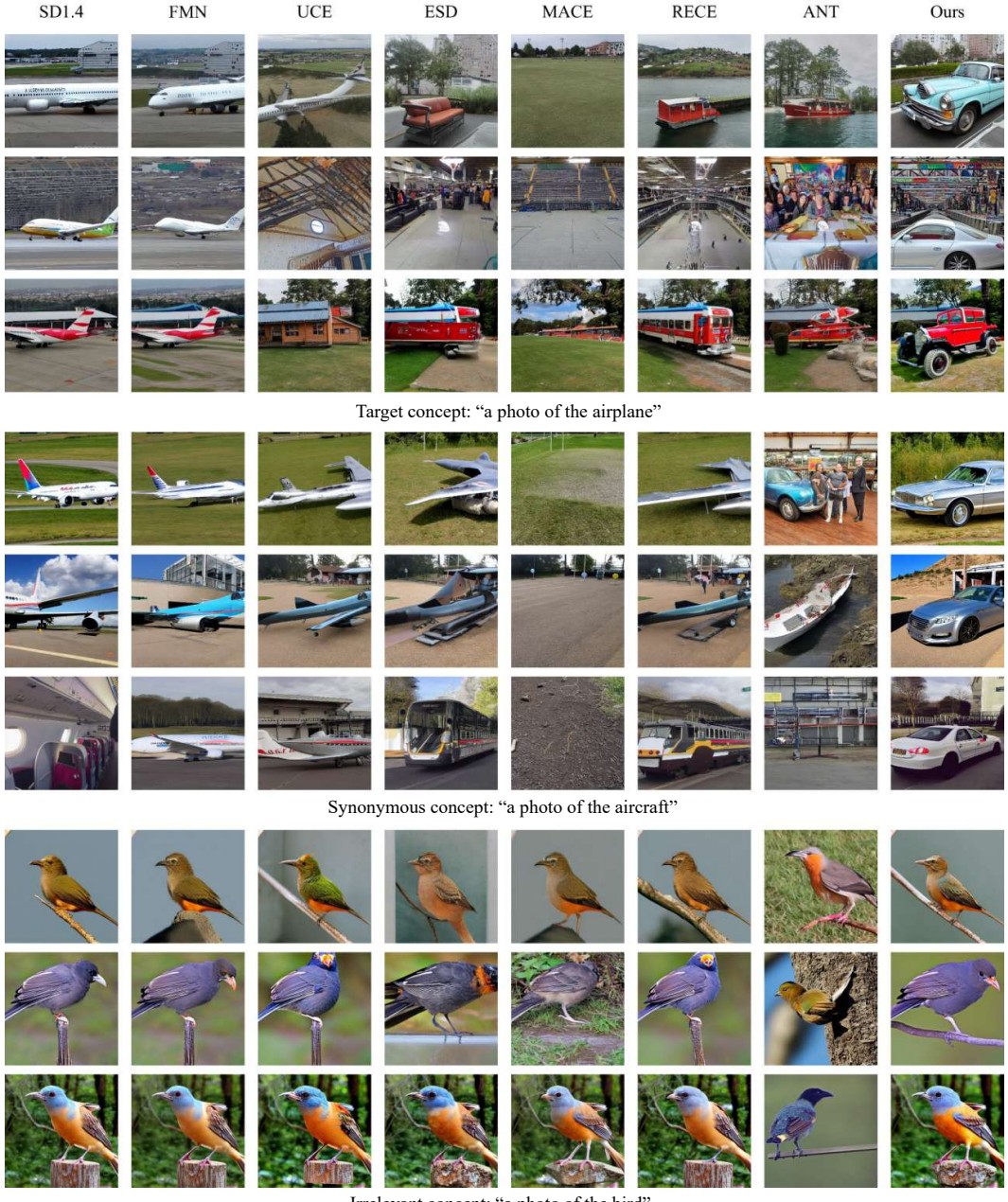

Figure 13: Qualitative results on airplane erasure. The images on the same row are generated using the same random seed.

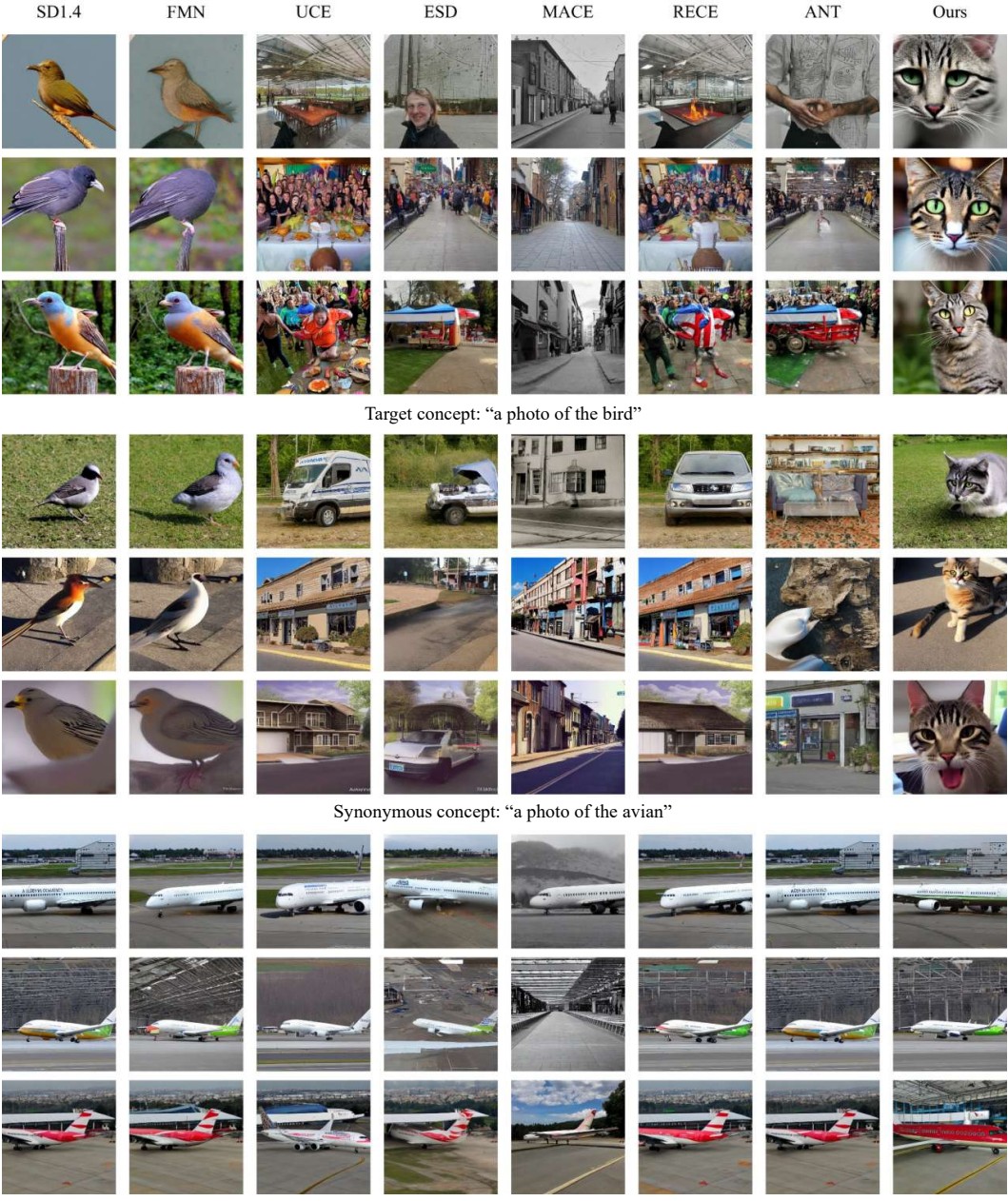

SD1.4    FMN    UCE    ESD    MACE    RECE    ANT    Ours

Target concept: "a photo of the bird"

Synonymous concept: "a photo of the avian"

Irrelevant concept: "a photo of the airplane"

Figure 14: Qualitative results on automobile erasure. The images on the same row are generated using the same random seed.

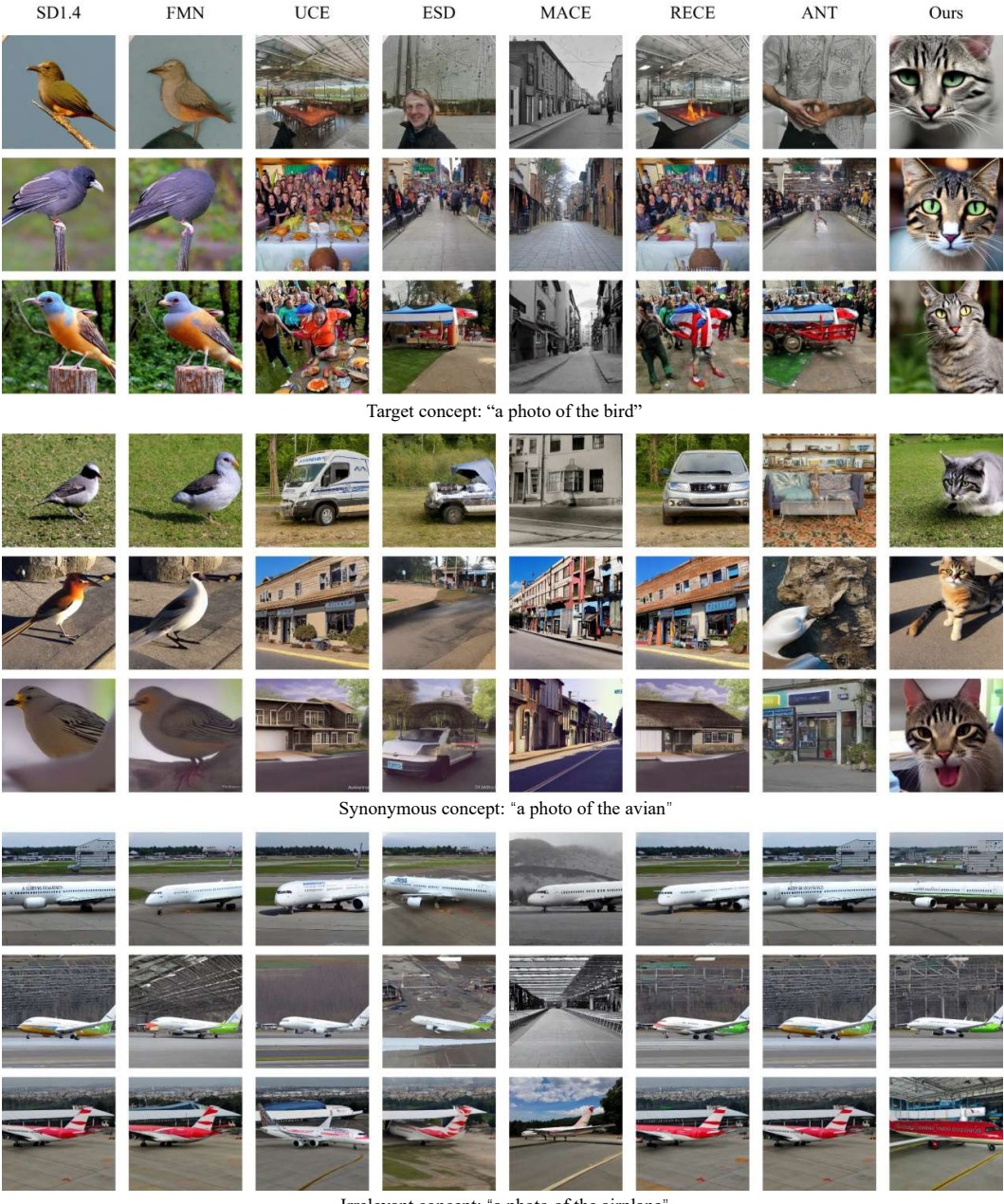

Figure 15: Qualitative results on bird erasure. The images on the same row are generated using the same random seed.

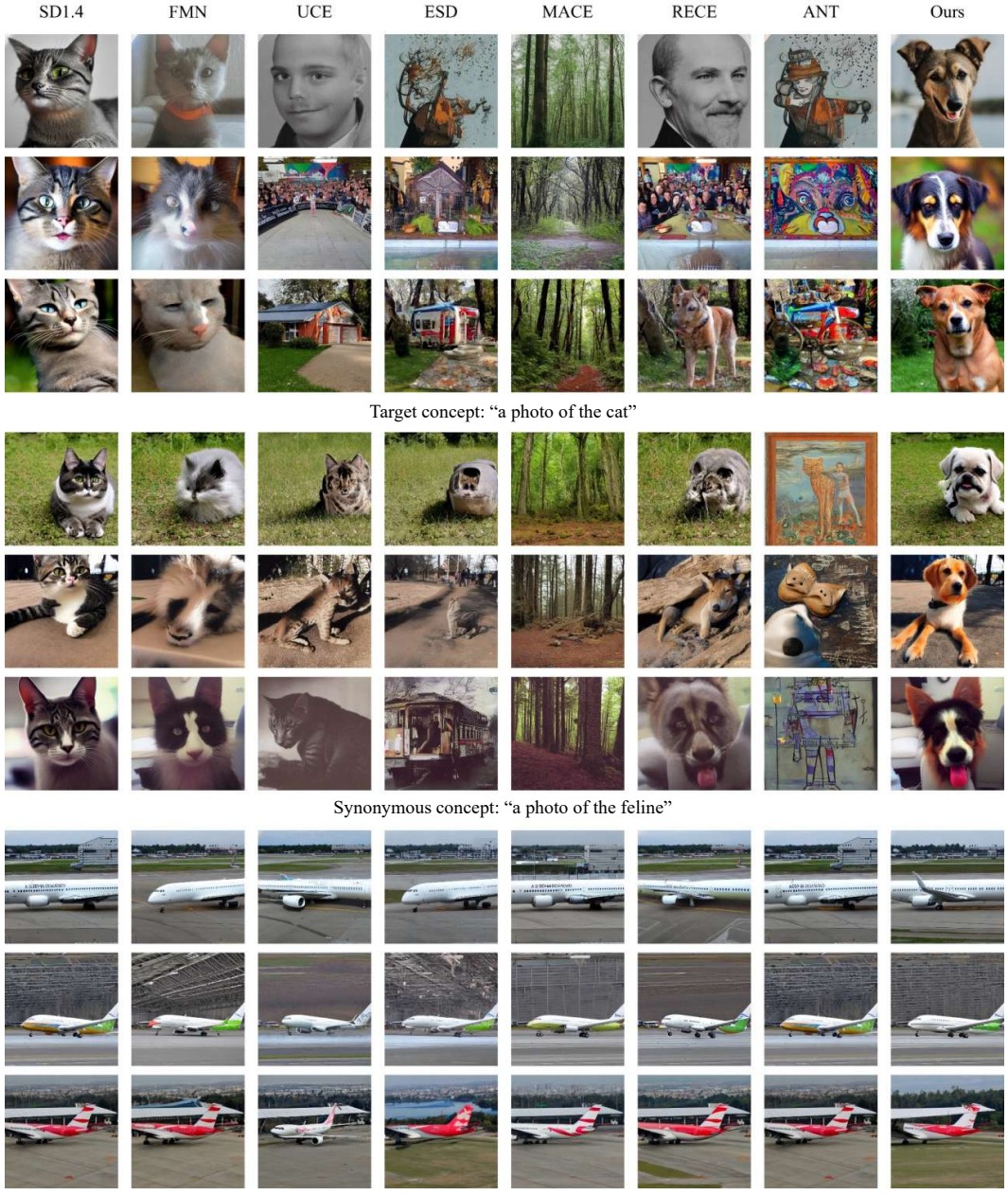

Figure 16: Qualitative results on cat erasure. The images on the same row are generated using the same random seed.

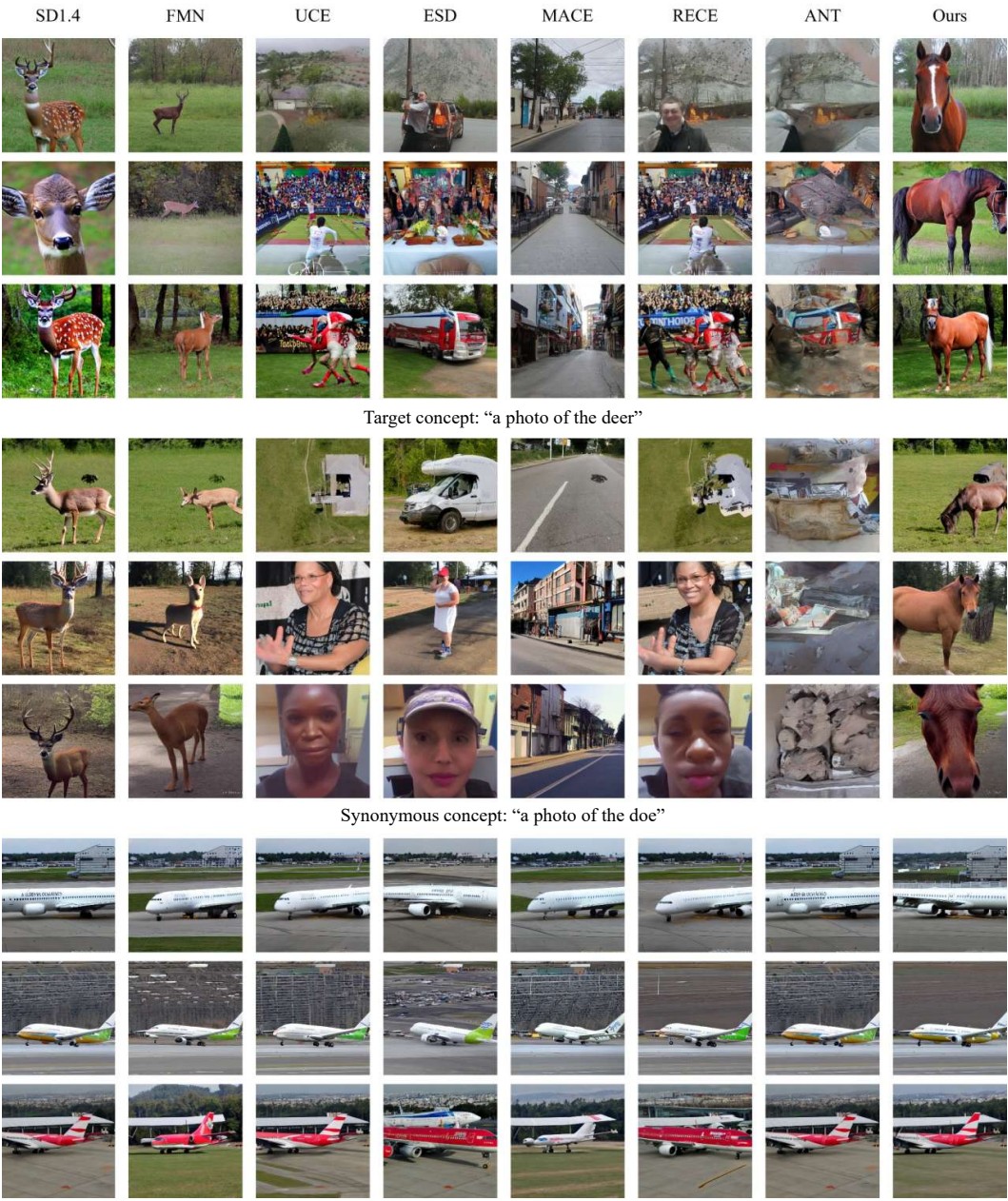

Figure 17: Qualitative results on deer erasure. The images on the same row are generated using the same random seed.

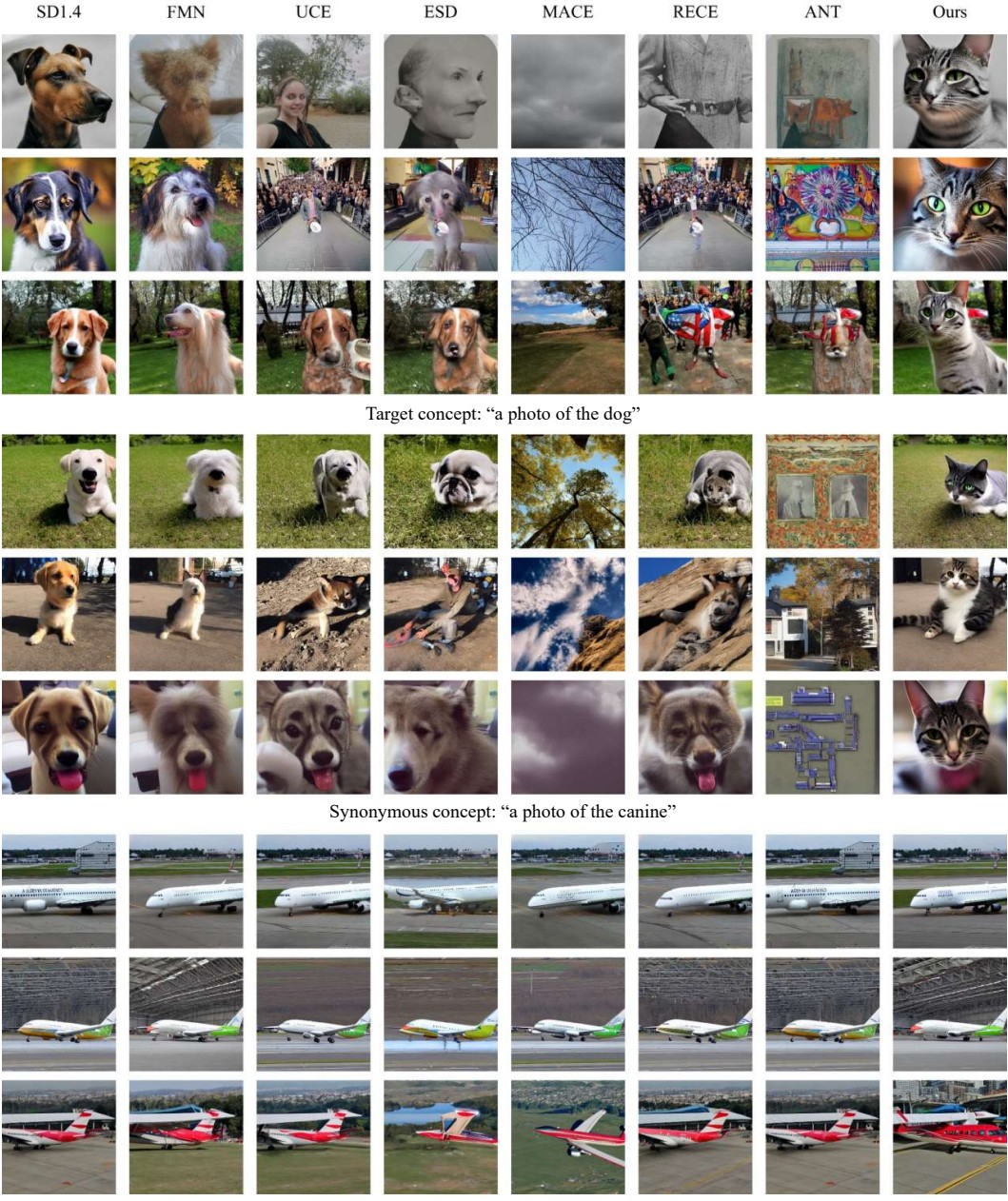

Figure 18: Qualitative results on dog erasure. The images on the same row are generated using the same random seed.

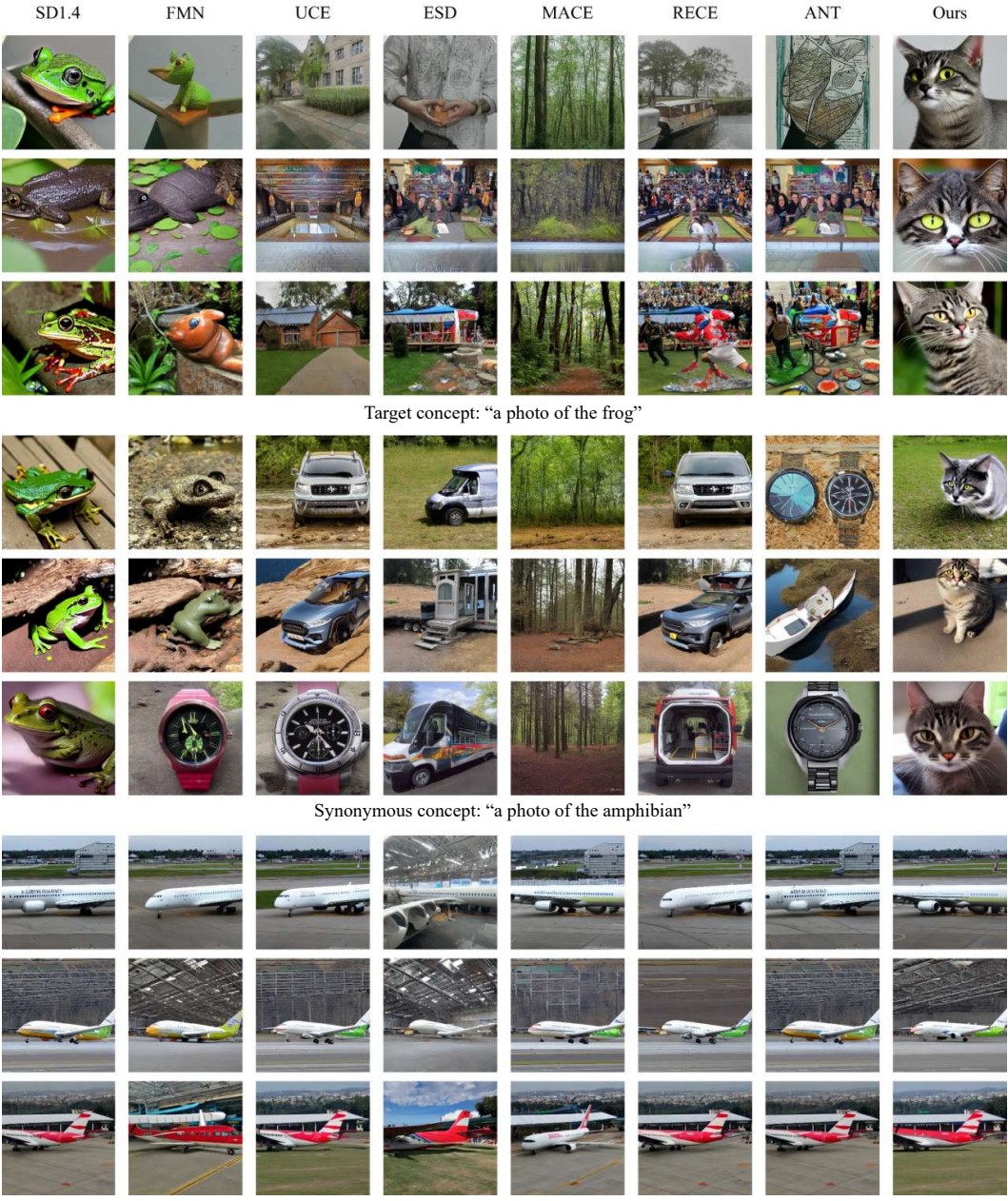

Figure 19: Qualitative results on frog erasure. The images on the same row are generated using the same random seed.

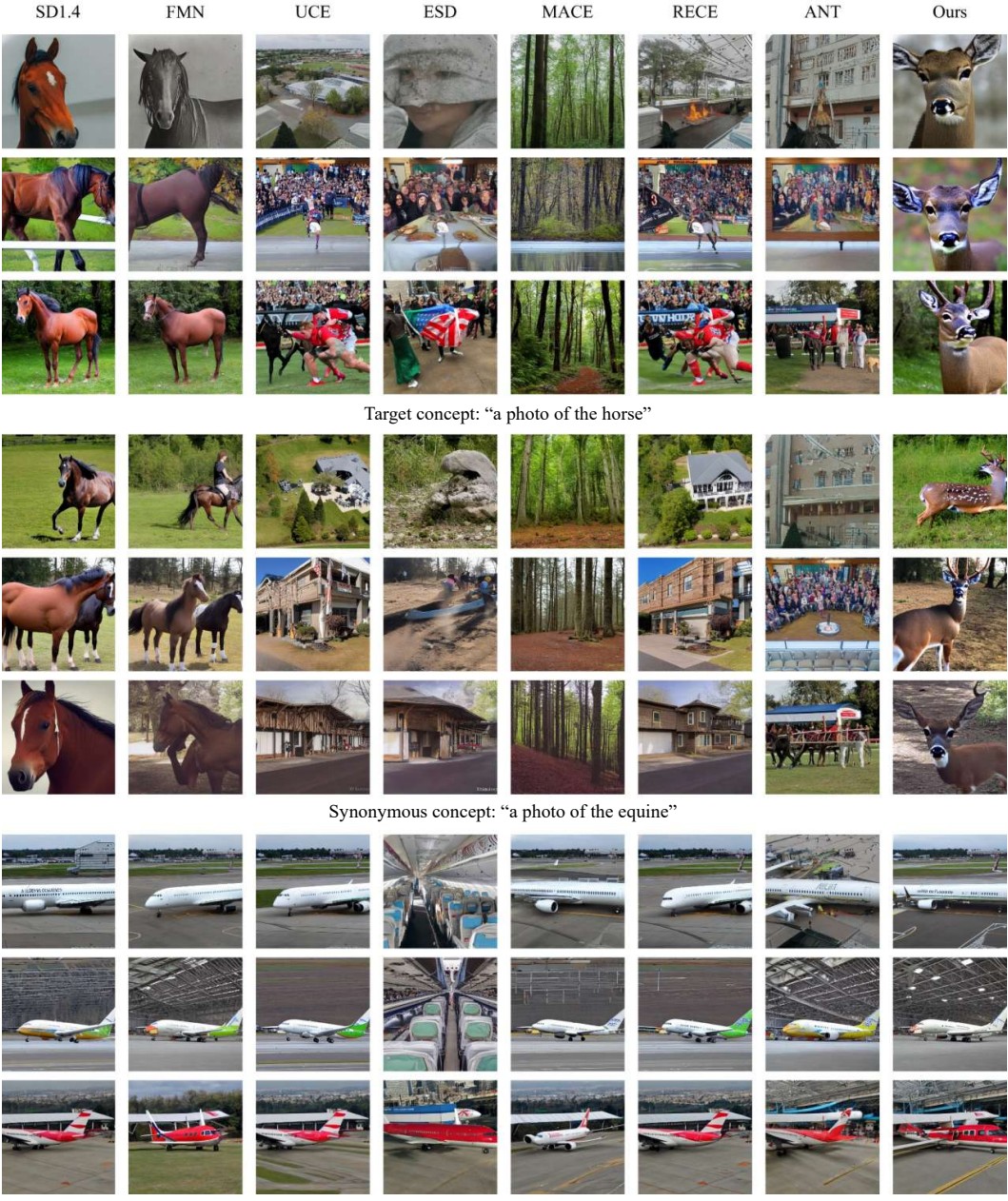

Figure 20: Qualitative results on horse erasure. The images on the same row are generated using the same random seed.

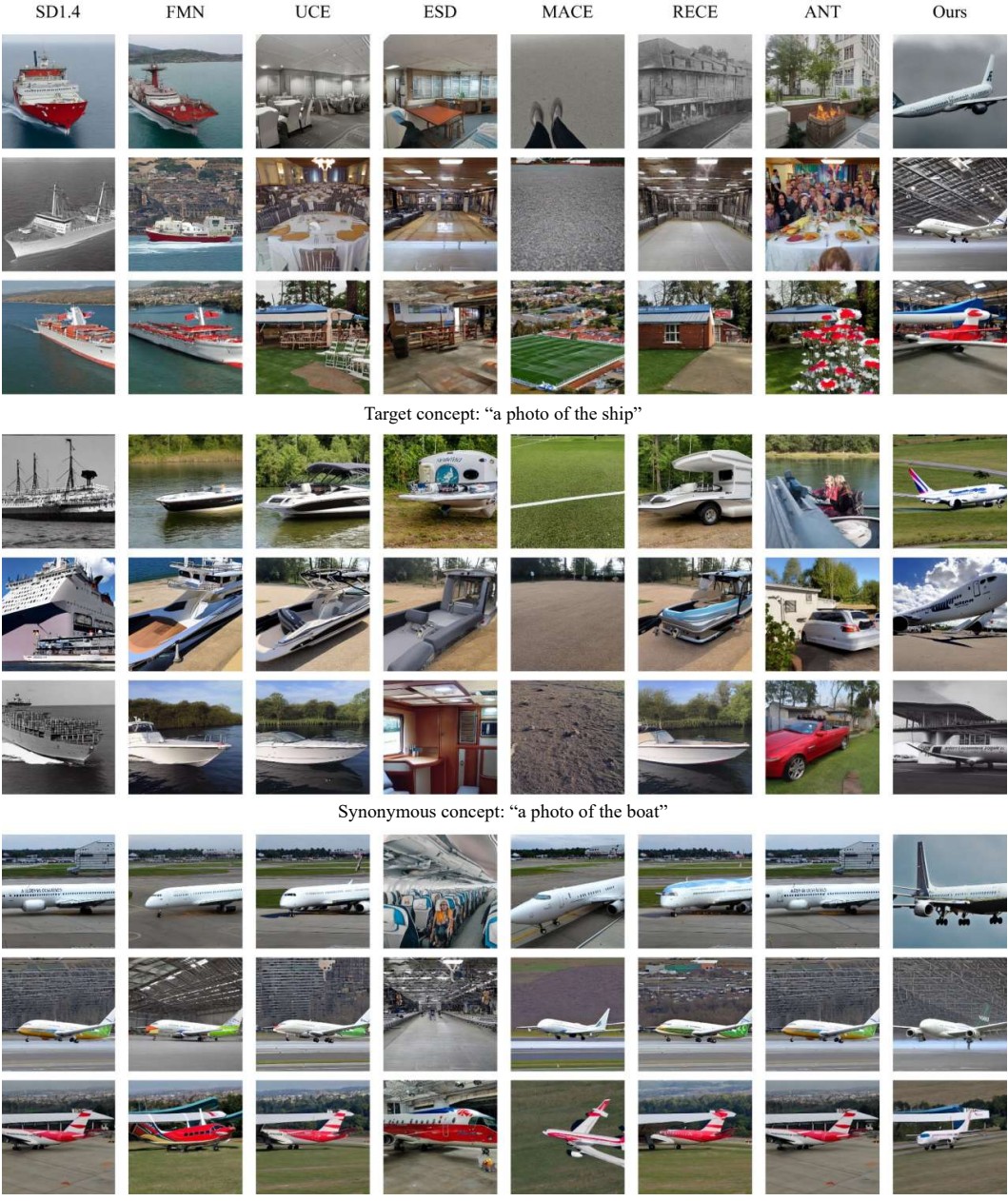

Figure 21: Qualitative results on ship erasure. The images on the same row are generated using the same random seed.

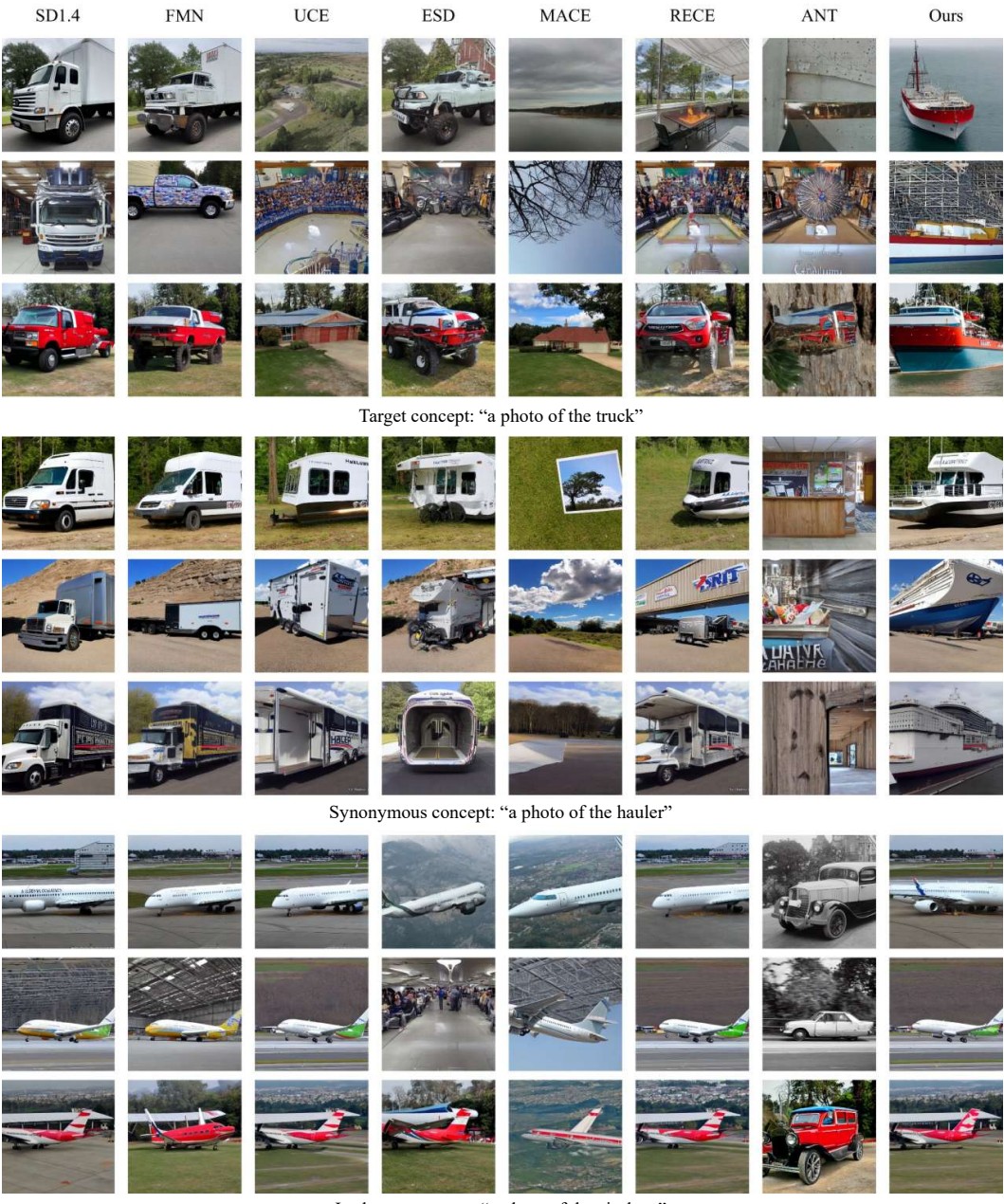

Figure 22: Qualitative results on truck erasure. The images on the same row are generated using the same random seed.

