# OpenReview forum: "SDErasure: Concept-Specific Trajectory Shifting for Concept Erasure via Adaptive Diffusion Classifier"
_ICLR.cc/2026/Conference — ICLR 2026 Poster_

### Official Review · Reviewer_DdAx · 2025-10-26

**Soundness:** 2
**Presentation:** 3
**Contribution:** 2
**Rating:** 4
**Confidence:** 4

**Summary:**

This paper observes that different concepts show distinct generative processes, with critical timesteps varying across concepts. Building on this insight, the paper introduces SDErasure, a novel concept erasure framework for text-to-image diffusion models. First, it proposes a step selection strategy to identify the critical timesteps most relevant to the target concept. It then introduces a score rematching loss to steer the generative process away from the target concept and toward an anchor concept.
To preserve the model’s generative capability, two additional regularization losses are proposed: the early-preserve loss, which enforces consistent noise predictions for target concepts at early steps, and the concept-retain loss, which maintains prediction consistency for non-target concepts after erasure.
Comparisons with prior concept erasure methods across four different domains (objects, celebrities, artistic styles, and sensitive content) demonstrate the effectiveness of SDErasure.

**Strengths:**

- The idea of adaptively selecting critical timesteps for concept erasure is novel and well-motivated.
- The proposed framework is efficient and effective, achieving strong performance in erasing target concepts while preserving non-target ones in all settings.
- The paper is well-written and easy to follow.

**Weaknesses:**

- The paper lacks an ablation study on the timestep selection strategy. It is unclear how the identified timesteps are verified to be optimal. For example, in Figure 6, the critical timesteps for erasing the Van-Gogh-style concept are found in the mid-to-late stages, but the truly optimal timesteps might be in the early-to-middle stages.
- The generalizability of the step selection strategy remains uncertain. If the total number of timesteps changes (e.g., from 50 to 200), can the same strategy still identify the correct critical timesteps?
- Since SSScore plays a crucial role in SDErasure, it would be valuable to include additional analyses similar to Figure 6. For example, what are the SSScore distributions for different concept categories, such as objects and celebrities? Do they follow consistent patterns, e.g., critical timesteps for objects appearing in mid stages and for celebrities in mid-to-late stages?
- In the multi-concept erasure experiments (Table 10), several established methods, such as UCE, MACE, and RECE, are omitted. Including these baselines would provide a more complete evaluation.
- (minor) In Appendix E, what do the hyper-parameters $\alpha$ and $\beta$ represent? Are they typos for $\beta_1$​ and $\beta_2$​ in Equation (14)?
- (minor) What values are used for the hyper-parameter $\eta$ in Equation (11) and for the iteration count $K$ in the pseudocode?

**Questions:**

See the weaknesses above.

---

> ### Author Response · Authors · 2025-11-21
> **Response to Reviewer DdAx  Part (1/2)**
>
> Thank you for your detailed and insightful review of our paper. We have uploaded a revised version of the manuscript, and all references in this comment correspond to the updated version. We appreciate the opportunity to address your concerns with clarifications from our paper and new comparative data.
>
> > **W1:** The paper lacks an ablation study on the timestep selection strategy. It is unclear how the identified timesteps are verified to be optimal.
>
> Thank you for raising this important point. We have added an additional ablation study in the revised version to verify that the identified timesteps are indeed optimal. Specifically, we divide all timesteps [1, 49] into ten equal groups: Group 1 corresponds to steps [45, 49], and Group 10 corresponds to steps [1, 5]. For each group, we train a model using only the timesteps in that group to erase the Picasso style, and then evaluate both the erasure effectiveness (CLIP) and the general generation quality (FID).
>
> To make the results more intuitive, we plot the model performance as a function of the group index in Appendix E, Figure 8 (left), along with the corresponding SSScore bar chart (right). The results clearly show that the best erasure performance is achieved by Group 5 (lowest CLIP), whose timesteps fall within steps [25, 30], precisely the region where the SSScore attains its maximum.
> The detailed results are provided in the following table. Although Group 5 achieves the strongest erasure performance among all groups, its FID remains relatively high, indicating a noticeable degradation in generation quality. In contrast, our method (Ours), trained at the identified optimal timesteps and equipped with Quality Regulation, attains strong erasure while significantly improving generation quality. Overall, it delivers the best balance between concept removal and visual fidelity.
>
> | | G1 | G2 | G3 | G4 | G5 | G6 | G7 | G8 | G9 | G10 | Ours |
> | :--- | :---: | :---: | :---: | :---: | :---: | :---: | :---: | :---: | :---: | :---: | :---: |
> | **CLIP**$\downarrow$ | 23.23 | 20.70 | 20.99 | 20.38 | $\underline{20.22}$ | 20.40 | 20.71 | 21.15 | 21.80 | 22.61 | $\textbf{19.55}$ |
> | **FID**$\downarrow$ | 14.01 | 14.81 | 14.48 | 14.33 | 13.98 | 13.90 | 13.27 | 12.73 | 11.96 | $\textbf{11.40}$ | $\underline{11.75}$ |
>
> > **W2:** The generalizability of the step selection strategy remains uncertain. If the total number of timesteps changes (e.g., from 50 to 200), can the same strategy still identify the correct critical timesteps?
>
> We appreciate the reviewer’s insightful observation. The answer is yes, The SSScore identifies the **relative generative phase** (noise level), not the absolute index. We present the SSScore predictions for total timesteps of 50, 100, and 200 in Appendix F, Figure 9. As shown in the figure, despite changes in the total number of steps, the predicted optimal denoising phase remains consistent. This demonstrates that SSScore identifies the relative generative phase rather than relying on the absolute timestep index.

---

> ### Author Response · Authors · 2025-11-21
> **Response to Reviewer DdAx  Part (2/2)**
>
> > **W3:** Since SSScore plays a crucial role in SDErasure, it would be valuable to include additional analyses similar to Figure 6. For example, what are the SSScore distributions for different concept categories?
>
> Thank you for the valuable suggestion. Including additional analyses similar to Figure 6 indeed helps better illustrate the role of SSScore. We have incorporated these results and added more SSScore bar plots for various categories in Appendix E (Figure 7).
>
> As shown in the expanded analysis, the critical timesteps for structural concepts (e.g., objects) consistently emerge in the early–mid denoising stages, whereas the critical timesteps for fine-grained concepts (e.g., celebrities) appear in the mid–late stages. These results further support our observations about the distinct temporal characteristics of different concept types.
>
> > **W4:** In the multi-concept erasure experiments (Table 10), several established methods, such as UCE, MACE, and RECE, are omitted. Including these baselines would provide a more complete evaluation.
>
> This is an excellent suggestion. We omitted them initially but we have now conducted a direct comparison. We evaluate UCE, MACE, and ANT on multi-concept erasure using the same experimental setup as Appendix I.3. Since the official RECE repository does not provide hyperparameters for multi-concept erasure, we are unable to directly evaluate it. Therefore, we use the more recent method ANT as a substitute for comparison. Due to page limitations, we placed these experimental results in Appendix I.3.
>
> | **Method** | $Average~CLIP\downarrow$ | $FID\downarrow$ |
> | :--- | :---: | :---: |
> | UCE | 23.24 | 16.87 |
> | MACE | 21.35 | 14.90 |
> | ANT | 21.74 | 14.18 |
> | **Ours** | **17.46** | **8.82** |
> | Original | 28.56 | - |
>
> As shown in the table, our method achieves markedly better performance than previous approaches. The CLIP score improves from 21.35 to 17.46, and the FID improves from 14.18 to 8.82.
>
> These gains come from two factors. First, SSScore accurately identifies the key timesteps responsible for generating each target concept. This allows the erasure to focus on the data manifold of each concept and avoids interfering with unrelated concepts. Second, Quality Regulation further protects the overall generation quality of the model.
>
>
> > **W5 & W6 (Minor Issues):**
> > * (W5) In Appendix E, what do the hyper-parameters $\alpha$ and $\beta$ represent? Are they typos for $\beta_1$ and $\beta_2$ in Equation (14)?
> > * (W6) What values are used for the hyper-parameter $\eta$ in Equation (11) and for the iteration count $K$ in the pseudocode?
>
> Thank you for your careful observation and constructive remark.
> * **Typos in Appendix E (W5):** You are correct. The symbols $\alpha$ and $\beta$ in Appendix E are typos; they refer to $\beta_1$ and $\beta_2$ in **Equation 14**. We have corrected this in our new version.
> * **Hyperparameters (W6):**
>     * In Equation (11), $\eta$ is set to **1.0**.
>     * In Algorithm 1, the iteration count $K$ is set to **20**.
>
> We have thoroughly revised and expanded the manuscript based on your suggestions, which has made the paper substantially more complete. We sincerely appreciate your constructive feedback.

---

> ### Comment · Reviewer_DdAx · 2025-11-28
> **Official Comment by Reviewer DdAx**
>
> Thank you for addressing my concerns. I have one remaining suggestion: the analysis in Figure 8 is conducted only for the style concept. It would be more convincing if a similar analysis is also performed for object concepts.
>
> **I would like to raise my score to 6**, but currently OpenReview does not show the "edit" button, so I'm unable to change my rating.

---

> > ### Author Response · Authors · 2025-12-01
> > **Response to reviewer DdAx**
> >
> > We sincerely thank you for your positive feedback and for acknowledging the value of our rebuttal. We are greatly encouraged by your intention to **raise the score to 6**.
> >
> > **Regarding the suggestion on Figure 8**: We fully agree that extending the analysis in Figure 8 to include object concepts would make our empirical validation more convincing and comprehensive. We commit to conducting this additional analysis and incorporating the corresponding results into the final version of the manuscript.
> >
> > Thank you again for your constructive guidance, which has helped us significantly improve the quality of our paper.

---

### Official Review · Reviewer_drbX · 2025-10-30

**Soundness:** 2
**Presentation:** 2
**Contribution:** 3
**Rating:** 4
**Confidence:** 4

**Summary:**

This paper presents SDErasure, a new framework for concept-specific erasure in text-to-image (T2I) diffusion models. Rather than modifying the full model, SDErasure focuses on selectively fine-tuning critical denoising timesteps, which are identified using a Step Selection algorithm. This approach allows the model to effectively erase target concepts with minimal disruption to unrelated content. By introducing two key technique: Score Rematching Loss and Quality Regulation, the method achieves a strong balance between effective concept erasure and preserving generative quality. Experimental results show that SDErasure outperforms existing methods in various scenarios, especially when erasing semantically meaningful or fine-grained concepts.

**Strengths:**

1. The paper makes a key observation: only a subset of denoising timesteps are critical for concept representation in diffusion models. By fine-tuning these specific steps instead of the whole model, the method is both efficient and targeted, reducing unnecessary side effects.
2. Quality Regulation is an important contribution. It explicitly addresses the trade-off between removing unwanted concepts and maintaining high-quality generations. This helps prevent issues like over-erasure or degradation of unrelated content.
3. The use of Score Rematching Loss aligns the denoising trajectory after erasure with the expected distribution.

**Weaknesses:**

1. The method’s effectiveness appears to be sensitive to anchor concept selection. This dependence might limit the method's generalizability across concepts that lack well-defined or semantically close anchors. A more robust or anchor-independent strategy would improve usability.

2. While the Step Selection algorithm is central to the method, the paper does not provide a theoretical justification for why the selected timesteps are optimal or what properties make certain steps more important. This makes it harder to understand or predict the method's behavior.

3. Although the method performs well on many tasks, performance gains are inconsistent. In some cases, improvements are small or even absent. The paper could benefit from a more in-depth analysis of failure cases, including explanations of when and why the method struggles.

4. Several related works on concept erasure and safety in T2I models are not cited or discussed. Notably missing are:

[1] Eraseanything: Enabling concept erasure in rectified flow transformers

[2] Dark miner: Defend against unsafe generation for text-to-image diffusion models

[3] One Image is Worth a Thousand Words: A Usability Preservable Text-Image Collaborative Erasing Framework

[4] Erasing More Than Intended? How Concept Erasure Degrades the Generation of Non-Target Concepts

**Questions:**

1. How sensitive is the method to the choice of threshold in the Step Selection algorithm? Does this threshold need to be manually tuned for each concept, or can it be set universally? Is there any theoretical or empirical basis for choosing an optimal value?

2. Structural concepts (e.g., human pose or object layout) are often deeply embedded in the generation process. Could the Quality Regulation mechanism interfere with the goal of erasing such structural concepts, especially if they strongly influence generation quality?

3. For multi-concept erasure, if both structural and fine-grained concepts need to be erased simultaneously, will the method still work effectively?

4. SDErasure effectively preserves the generative quality of unrelated concepts while removing the target concept. Can you provide a more detailed analysis of how Quality Regulation and Score Rematching loss impact generative quality, respectively?

---

> ### Author Response · Authors · 2025-11-21
> **Response to Reviewer drbX Part (1/4)**
>
> Thank you for your insightful comments and constructive criticism on our paper. We have uploaded a revised version of the manuscript, and all references in this comment correspond to the updated version. We have tried to resolve your concerns as follows:
>
> > **W1:** The method’s effectiveness appears to be sensitive to anchor concept selection. This dependence might limit the method's generalizability across concepts that lack well-defined or semantically close anchors.
>
> We apologize for not making our contributions clear enough. Our paper discussed anchor selection as a separate topic, which may have caused some misunderstanding. Our method is not sensitive to the choice of anchor. Hopefully, the following explanation will make it clearer.
>
> Our method is not dependent on anchors and most of our key results is operated in a completely anchor-free manner.
>
> * **Anchor-Free is the Default:** Our method is designed to support anchor-free erasure. For all fine-grained concepts, such as celebrities and artistic styles, we default to and recommend this anchor-free approach. The SOTA results we present in Table 2 and Table 3 were all achieved in this anchor-free setting.
> * **Anchors are Optional:** Anchors are merely an optional tool used primarily when erasing structural concepts like objects, where they can help guide the generative trajectory to a similar, valid manifold.
> * **Robustness to Anchor Choice:** Even when anchors are used, our method is highly robust. We provide a detailed ablation in Appendix H.3 (Table 10). This table explicitly compares our heuristic selection ("Ours") against "Empty" (anchor-free) and "Random" (considered as suboptimal) anchors. The results show that even when using a sub-optimal anchor (random), our method still achieves performance superior ($H_0 = 90.21$%) to all baselines except MACE. Although slightly lower than MACE, it is important to note that MACE requires images containing the target concept along with SAM-generated masks as training data, whereas our method attains comparable effectiveness using only a single prompt.
>
> This proves that SDErasure's core advantages stem from Step Selection and Quality Regulation, not from anchors. The method is broadly generalizable and user-friendly.
>
> Prior methods such as UCE, MACE, and RECE also involve the choice of anchors, and different anchors may lead to slight performance variations. We clarify that just like previous methods, the performance of SDErasure primarily comes from the core algorithmic design rather than the specific choice of anchors.
>
> We have revised the paper accordingly. In the new version, we added comparisons and analyses of different anchor selection strategies in the ablation study section.

---

> ### Author Response · Authors · 2025-11-21
> **Response to Reviewer drbX Part (2/4)**
>
> > **W2:** While the Step Selection algorithm is central to the method, the paper does not provide a theoretical justification for why the selected timesteps are optimal or what properties make certain steps more important.
>
> We thank the reviewer for this insightful comment. We clarify that our Step Selection strategy is not merely heuristic but is grounded in both the **geometric properties of the score function** and the **probabilistic framework of the Evidence Lower Bound (ELBO)**.
> We have incorporated the following theoretical derivation into the **Method section** of the revised manuscript.
>
> #### 1. Geometric Motivation: Score Function Divergence
> First, we characterize the direction of generation using the score function, defined as the gradient of the log-density. At each timestep $t$, the generative directions conditioned on the target concept $c_t$ and the anchor concept $c_a$ are given by:
> $$g_t^{(c)} = \nabla_{x_t}\log p_\theta(x_t \mid c_t), \quad g_t^{(a)} = \nabla_{x_t}\log p_\theta(x_t \mid c_a).$$
>
> To quantify the divergence between these directions, we leverage the relationship between the score function and the noise prediction network in diffusion models [1]. Specifically, the score function is approximated by the scaled noise prediction:
> $$\nabla\_{x\_t}\log p\_\theta(x\_t \mid c) \approx -\frac{1}{\sqrt{1-\bar{\alpha}\_t}}\epsilon\_\theta(x\_t, t, c).$$
>
> This equivalence allows us to reformulate the gradient divergence directly in terms of the model's noise prediction errors. The discrepancy for the target and anchor concepts is measured as:
> $$\mathcal{L}\_{t}^{(c)} = \bigl\|\,\epsilon\_\theta(x\_t, t, c\_t) - \epsilon\bigr\|\_2^2, \quad \mathcal{L}\_{t}^{(a)} = \bigl\|\,\epsilon\_\theta(x\_t, t, c\_a) - \epsilon\bigr\|\_2^2.$$
>
> **Geometrically**, a large difference between $\mathcal{L}\_{t}^{(a)}$ and $\mathcal{L}\_{t}^{(c)}$ implies that the vector field guiding the target concept points in a significantly different direction than that of the anchor, indicating a critical bifurcation in their generative trajectories.
>
> #### 2. Probabilistic Quantification: Instantaneous Posterior
> While the geometric perspective highlights the presence of trajectory shifts, it lacks a normalized metric for automated selection. To strictly quantify this divergence, we interpret the losses through the lens of Diffusion Classifiers [2]. The fundamental principle explaining why diffusion models function as classifiers has been discussed in [2].
>
> * **Log-Likelihood Approximation:** The noise prediction error approximates the negative log-likelihood of the data:
>     $$\log p(x\_t \mid c) \approx -\mathcal{L}\_t^{(c)} + C.$$
> * **SSScore Derivation:** Consequently, the term $\exp(-\mathcal{L}\_{t}^{(c)})$ serves as a proxy for the unnormalized instantaneous likelihood. Based on Bayes' theorem (assuming uniform priors), we formulate the SSScore as the **instantaneous posterior probability**:
>     $$\mathcal{S}\_t = \frac{p(c\_t \mid x\_t)}{p(c\_t \mid x\_t) + p(c\_a \mid x\_t)} \approx \frac{\exp(-\mathcal{L}\_{t}^{(c)})}{\exp(-\mathcal{L}\_{t}^{(c)})+\exp(-\mathcal{L}\_{t}^{(a)})}.$$
>
> #### 3. Optimal Intervention Point
> Combining these perspectives, a high $\mathcal{S}\_t$ signifies that: (1) **Geometrically**, the generative trajectory of the target concept has decoupled from the anchor's trajectory, and (2) **Probabilistically**, the model holds high confidence that it is synthesizing features unique to the target. This identifies timestep $t$ as a **critical intervention point** to perform precise erasure with minimal collateral damage to the shared generative manifold.
>
> #### 4. Empirical Validation of Theory.
> We provide empirical proof that this theoretical metric correctly identifies concept-specific generative dynamics in Appendix E (Figure 6):
> * **Van Gogh Style,**  characterized by high-frequency details like rich brushstrokes, SSScore correctly identifies mid-to-late timesteps where fine details gradually appear.
> * **Picasso Style,** characterized by low-frequency structural elements such as large color blocks, SSScore correctly identifies early-to-mid steps where global structure is formed.
>
> In summary, the selected timesteps are optimal because they correspond to the moments of maximum probabilistic separability, ensuring the erasure is strictly confined to the target concept's unique generative path.

---

> ### Author Response · Authors · 2025-11-21
> **Response to Reviewer drbX Part (3/4)**
>
> > **W3:** Although the method performs well on many tasks, performance gains are inconsistent. In some cases, improvements are small or even absent. The paper could benefit from a more in-depth analysis of failure cases.
>
> Thanks for raising this issue. We argue that the core contribution of our method is to greatly improve **generation quality** while ensuring the erasure capability.
> That is to say, compared with Fine-tuning based methods (e.g., ESD, which often significantly degrade the model's generation quality), we consistently outperform in FID.
> While compared with closed-form solution methods (e.g., MACE, RECE, SPEED, which tend to result in incomplete erasure), we are superior in CLIP score (erasure capability).
> SDErasure allows us to achieve the optimal balance, and the performance gains are highly consistent when viewed through this lens.
>
> In Table 2 (celebrity erasure), although our method does not obtain the lowest CLIP score, the visualizations in Appendix Figure 12 show that the target concept is effectively removed. More importantly, our method reaches a substantially lower FID than prior approaches. Compared with the previous SOTA ANT (average FID of 12.47), our method achieves an FID of 7.27. This indicates that our approach attains a much better balance between concept removal and the preservation of non-target attributes.
>
> In Table 3 (artist-style erasure), although prior closed-form solution methods such as RECE, MACE and SPEED report relatively low FID scores, they all fail to fully erase at least one concept (CLIP > 23). In contrast, our method consistently achieves complete and stable erasure across all concepts while maintaining strong generative quality.
>
> Overall, these results show that previous methods struggle to balance erasure strength with preservation quality. SDErasure effectively addresses this issue by ensuring robust erasure without over-suppressing unrelated concepts. Although improvements on some individual metrics may appear modest, SDErasure provides the most stable and reliable performance across all metrics and concepts.
>
> > **W4:** Several related works on concept erasure and safety in T2I models are not cited or discussed.
>
> Thank you for highlighting these omissions. We have added a thorough discussion and citation for each of them in the related work section of our new version to provide a more comprehensive view of the current research landscape.
>
> ---
>
> > **Q1:** How sensitive is the method to the choice of threshold in the Step Selection algorithm? Does this threshold need to be manually tuned for each concept, or can it be set universally?
>
> We analyzed the impact of the threshold $\lambda$ in detail in our ablation study, Figure 5.
> The results show that our method is not highly sensitive to the choice of $\lambda$. A wide range of values between $[0.5, 0.8]$ yields a favorable trade-off between erasure strength and generation quality.
>
> Importantly, this threshold does not need to be tuned for each concept. We adopt a universal choice of $\lambda=0.8$ (based on the ablation study) for most experiments and $\lambda=0.5$ for object erasure, demonstrating the robustness of our approach. All experiments in this paper are thus conducted with a single global $\lambda$, further confirming that the method remains stable without requiring per-concept hyperparameter adjustment.
>
> > **Q2:** Structural concepts (e.g., human pose or object layout) are often deeply embedded in the generation process. Could the Quality Regulation mechanism interfere with the goal of erasing such structural concepts?
>
> This is an excellent and perceptive question. The answer is no, it does not interfere. Quality Regulation consists of two components: *early-preserve* and *concept-retain*. We explain below why neither of these components interferes with the erasure process.
>
> The *early-preserve* term is applied only to the first 3–5 denoising steps. As shown in the SSScore bar charts in Appendix E (Figure 7), all concepts exhibit consistently low SSScores during these early steps, indicating that their trajectories are highly similar at the beginning of denoising. Since erasure training primarily occurs after this stage, applying quality regulation here merely ensures that the generated samples remain on the manifold of natural images, without conflicting with the main objective of concept erasure.
>
> The goal of *concept-retain* is to preserve the model's generation capability for concepts unrelated to the target one. It is trained on data that does not contain the target concept. Therefore, it does not impede the removal of the target concept.

---

> ### Author Response · Authors · 2025-11-21
> **Response to Reviewer drbX Part (4/4)**
>
> > **Q3:** For multi-concept erasure, if both structural and fine-grained concepts need to be erased simultaneously, will the method still work effectively?
>
> This is a very insightful question. The answer is yes, it will work effectively. This is a key strength of our adaptive framework.
> If the task is to erase structural and fine-grained simultaneously, our Step Selection algorithm will identify two distinct sets of critical timesteps:
> a set of early-stage timesteps (for structural concepts) and
> a set of mid-to-late-stage timesteps (for fine-grained concepts).
> Our total loss $\mathcal{L}_o$ will then be applied to this combined set of timesteps. Because the interventions are targeted at different, largely non-overlapping phases of the generative process, the interference between the two erasure tasks is minimal, allowing for effective simultaneous removal.
>
> To better address this concern, we conducted a comprehensive experiment.
>
> We selected three representative fine-grained concepts ("Elon Musk," "Taylor Swift," and "Donald Trump") and three structural concepts ("airplane," "automobile," and "bird"). Both sets of concepts were co-erased simultaneously within a single model. The evaluation metrics employed for both the fine-grained and structural concepts remain consistent with those detailed in the main paper. The experimental results are presented in the following table.
>
> | **Erasing Fine-Grained Concepts** | | | |
> | :--- | :---: | :---: | :---: |
> | **Concept** | $CLIP_{ref}$ | $CLIP$ | $\Delta_{clip}\downarrow$ |
> | Elon Musk | 26.80 | 16.39 | 10.41 |
> | Taylor Swift | 24.94 | 16.67 | 8.27 |
> | Donald Trump | 23.57 | 16.94 | 6.63 |
> | **FID** | | **15.8** | |
>
> | **Concept** | **$Acc_e\downarrow$** | **$Acc_g\uparrow$** | **$Acc_s\downarrow$** | **$H_0\uparrow$** |
> | :--- | :---: | :---: | :---: | :---: |
> | Airplane | 1.53 | 96.17 | 1.97 | 97.54 |
> | Automobile | 2.34 | 96.17 | 15.5 | 92.39 |
> | Bird | 0.80 | 96.17 | 5.32 | 96.65 |
>
> From the table, we observe that for *celebrity concept erasure*, SDErasure substantially reduces the CLIP similarity between the generated images and their corresponding textual prompts. For *object-structure erasure*, SDErasure achieves consistently high $H_0$ scores, demonstrating its effectiveness in jointly removing both structural and fine-grained concepts.
>
> > **Q4:** SDErasure effectively preserves the generative quality of unrelated concepts while removing the target concept. Can you provide a more detailed analysis of how Quality Regulation and Score Rematching loss impact generative quality, respectively?
>
> We appreciate the opportunity to address your concerns.
> Below is a detailed analysis of how Quality Regulation and Score Rematching loss impact generative quality.
>
> **Quality Regulation.** Quality Regulation includes two components, *early-preserve* and *concept-retain*. They protect the model's generation quality from two different aspects.
> * **Early-preserve.** The early denoising steps mainly guide Gaussian noise toward the natural image manifold. If this stage is disturbed, the generated images may drift away from the natural distribution and show artifacts. Early-preserve keeps the early denoising trajectory of the target concept unchanged. This prevents large deviations in the erased concept's trajectory.
> * **Concept-retain.** This term has a different goal. It encourages the model to match the noise predictions of the original model on a wide range of unrelated concepts. In this way, the model maintains its general generation quality.
>
> We also provide a precise quantitative analysis of these components in our ablation study, Table 5.
>
> **Score Rematching ($\mathcal{L}_e$):** This is the primary erasure mechanism. Using it alone ("Baseline" in Table 5) achieves erasure but results in a higher FID of **8.05**.
>
> **Quality Regulation ($\mathcal{L}_r$ and $\mathcal{L}_p$):** These are the primary quality *preservation* mechanisms.
> * Adding only $\mathcal{L}_r$ (concept-retain) to the baseline improves the FID from 8.05 to **7.39**. This shows $\mathcal{L}_r$ effectively preserves unrelated concept fidelity.
> * Adding only $\mathcal{L}_p$ (early-preserve) to the baseline improves the FID from 8.05 to **7.06**. This shows $\mathcal{L}_p$ is highly effective at preserving the model's structural integrity.
> * Adding both ("Ours") achieves the final, lowest FID of **6.65**.
>
> **Conclusion:** Score Rematching ($\mathcal{L}_e$) is responsible for erasure, while the two components of Quality Regulation ($\mathcal{L}_p$ and $\mathcal{L}_r$) are demonstrably responsible for preserving generation quality along two dimensions.
>
> Reference:
>
> [1] Ho, Jonathan, Ajay Jain, and Pieter Abbeel. "Denoising diffusion probabilistic models." Advances in neural information processing systems 33 (2020): 6840-6851.
>
> [2] Li, Alexander C., et al. "Your diffusion model is secretly a zero-shot classifier." Proceedings of the IEEE/CVF International Conference on Computer Vision. 2023.

---

### Official Review · Reviewer_tk2R · 2025-10-30

**Soundness:** 2
**Presentation:** 3
**Contribution:** 3
**Rating:** 6
**Confidence:** 3

**Summary:**

This paper proposes a fine-grained loss for concept erasure. It targets key timesteps critical for each concept, which are automatically identified using a diffusion classifier. By incorporating constraints to preserve untargeted concepts, the method achieves effective concept erasure in experiments. The authors evaluate the approach against multiple baselines.

**Strengths:**

1. The fine-grained loss across different timesteps for each concept is noteworthy, and the corresponding early-preserve loss mitigates the instability issues of fine-tuning.
2. The selection of critical timesteps demonstrates potential for effective concept erasure while preserving untargeted concepts.
3. The authors compare their method with recent attacks, such as SPEED and ANT, showing that it achieves superior performance.

**Weaknesses:**

1. Appendix G.3 presents the performance of the proposed method in multi-concept settings. However, comparisons with other methods in these settings should be included to better demonstrate the scalability of the approach.

2. Does the step selection lead to error prediction? This could help determine whether all concepts follow the same principles during step selection.

3. The SSScore is calculated at each timestep. How does this impact computational efficiency?

**Questions:**

1. Comparison with other methods in multi-concept settings.
2. Analysis of error predictions in step selection.
3. Assessment of computational efficiency.

---

> ### Author Response · Authors · 2025-11-21
> **Response to Reviewer tk2R Part (1/3)**
>
> Thank you for insightful feedback. We have uploaded a revised version of the manuscript, and all references in this comment correspond to the updated version. We have tried to resolve your concerns as follows:
>
> > **W1:** Appendix G.3 presents the performance of the proposed method in multi-concept settings. However, comparisons with other methods in these settings should be included to better demonstrate the scalability of the approach.
> >
> > **Q1:** Comparison with other methods in multi-concept settings.
>
> Thank you for this valuable feedback. We recognize that only presenting SDErasure's performance in Appendix G.3（Appendix I.3 in the new version）without a direct comparison to baselines may leave the evaluation less complete. Therefore, we added additional comparative experiments, which are included in Appendix I.3 due to space limitations in the main paper.
>
> Following your valuable suggestion, we have conducted a new supplementary experiment during the rebuttal period to directly compare SDErasure against SOTA methods on the multi-concept erasure task.
>
> **Experimental Setup.** We selected the three strongest baselines applicable for multi-concept erasure: UCE, MACE and ANT. We set the same task as our Appendix I.3 (erasing 20 celebrity concepts in a single model). We measure the erasure efficiency (CLIP, lower is better) and general generative quality (FID, lower is better) on the MSCOCO-30k dataset post-erasure. The results of our experiments are shown in the table below.
>
> | **Method** | $Average~CLIP\downarrow$ | $FID\downarrow$ |
> | :--- | :---: | :---: |
> | UCE | 23.24 | 16.87 |
> | MACE | 21.35 | 14.90 |
> | ANT | 21.74 | 14.18 |
> | **Ours** | **17.46** | **8.82** |
> | Original | 28.56 | - |
>
> The table above compares the performance of SDErasure against three strong baselines (UCE, MACE, ANT) on a multi-concept erasure task. SDErasure is the only method that optimizes for both conflicting goals. It achieves the best erasure performance (lowest CLIP) while simultaneously maintaining the best image quality (lowest FID).
>
> SDErasure demonstrates particularly strong performance in multi-concept erasure tasks, primarily owing to its architectural design. SSScore precisely localizes facial concepts to the mid-to-late denoising steps, ensuring that different concepts are removed within their respective flow subspaces while avoiding interference caused by overlapping denoising trajectories at early timesteps. Meanwhile, the Quality Regulation module further preserves the early-stage denoising trajectories.

---

> ### Author Response · Authors · 2025-11-21
> **Response to Reviewer tk2R Part (2/3)**
>
> > **W2:** Does the step selection lead to error prediction? This could help determine whether all concepts follow the same principles during step selection.
> >
> > **Q2:** Analysis of error predictions in step selection.
>
> This is a profound question. We have also considered this issue in our preliminary manuscripts. The step selection reveals the key insight that different concepts do not follow the same generative principle. Although we cannot directly prove that Step Selection never leads to error predictions, its effectiveness is supported by extensive experimental results.
>
> As shown in Figure 5, a threshold $\lambda=0$ corresponds to uniform sampling across all timesteps, while larger values of $\lambda$ indicate that we select timesteps with higher confidence. We observe that as $\lambda$ increases, the generation quality (measured by FID) gradually improves. Notably, when $\lambda$ is in the range $[0.5, 0.8]$, the model significantly outperforms random sampling ($\lambda=0$). These results demonstrate that Step Selection effectively identifies the optimal timesteps, thereby validating its effectiveness.
>
> To further illustrate this, we divide all timesteps evenly into ten groups, where Group 1 corresponds to steps [45, 49] and Group 10 corresponds to steps [1, 5]. We perform erasure on each group and evaluate the performance. Appendix E, Figure 8 (left) shows the erasure performance curve across groups, while the right figure presents the SSScore predicted during the Step Selection phase. From the left plot, it is evident that the best erasure performance is achieved by Group 5, corresponding to timesteps [25, 30], which coincides with the peak of SSScore in the right plot. This further validates the effectiveness of Step Selection. Figure 7 in Appendix E presents the SSScore peaks for different categories of concepts. As shown, the peak positions vary across concepts, further demonstrating that different concepts follow distinct principles.
>
> To better illustrate that different concepts follow distinct principles during step selection. We provide direct evidence for this in Appendix E, Figure 6:
> * **"Van Gogh Style":** Characterized by rich, high-frequency brushstrokes. Our SSScore correctly identifies its critical timesteps in the middle-to-late stages, which is when the model synthesizes fine details.
> * **"Picasso Style":** Characterized by large, low-frequency structural color blocks. Our SSScore correctly identifies its critical timesteps in the early-to-middle stages, which is when the model forms the macro-structure.
>
> This proves that different concepts do not follow the same principle. Our Step Selection mechanism adaptively captures these unique "generative fingerprints", which is the source of our method's precision and a core strength of its design.

---

> ### Author Response · Authors · 2025-11-21
> **Response to Reviewer tk2R Part (3/3)**
>
> > **W3:** The SSScore is calculated at each timestep. How does this impact computational efficiency?
> >
> > **Q3:** Assessment of computational efficiency.
>
> This is a crucial practical question. The SSScore calculation is a one-time, pre-processing step and is not part of the training loop or inference. Moreover, compared with state-of-the-art methods such as ANT, our overall computational cost is reduced by *75%*.
>
> The computation of SSScore incurs only minimal overhead. As detailed in Appendix G.1, for object erasure on CIFAR-10, the entire process takes only about 20 minutes on a single RTX 4090, with the Step Selection algorithm consuming only less than 3 minutes. Our total training time is reduced by approximately **75%** compared to previous trajectory-aware methods like ANT while also achieving substantially better generation quality. For example, in the celebrity erasure task, the FID improves from 12.47 (ANT) to 7.27 with our method. Thus, the slight overhead of SSScore is outweighed by the significant gain in training efficiency.
>
> We further explores two practical acceleration strategies:
>
> * **Sparse Sampling.**
>     Since the SSScore curve is smooth over timesteps as show in Appendix E, Figure 6, it is unnecessary to evaluate every step. Computing the score every $3$–$5$ steps and interpolating the remaining values preserves both erasure effectiveness and generation quality while reducing the pre-computation time by up to $\sim 80$%. The detailed experimental results are provided in Appendix C, Table 7.
> * **Concept Clustering for Multi-Concept Erasure.**
>     Fine-grained concepts (e.g., faces or styles) and structural concepts emerge at different denoising stages. By clustering concepts using an LLM and letting concepts in the same cluster share a single SSScore computation, the overall complexity is reduced from $O(N_{\text{concepts}})$ to $O(N_{\text{clusters}})$. This strategy is already validated in our multi-concept erasure experiments in Appendix I.3, Table 12, which use shared SSScore profiles without compromising accuracy.
>
> Together, these two strategies demonstrate that SSScore not only introduces negligible overhead but can be computed even more efficiently in practice, further strengthening its practicality for large-scale erasure tasks.

---

### Official Review · Reviewer_DjSd · 2025-10-31

**Soundness:** 3
**Presentation:** 3
**Contribution:** 3
**Rating:** 6
**Confidence:** 5

**Summary:**

This paper introduces SDErasure, a concept erasure method for diffusion models that adaptively targets concept-specific timesteps based on a diffusion classifier. It proposes a Step Separability Score (SSScore) to select critical steps, and introduces Score Rematching and Quality Regulation to balance erasure efficacy and generation quality. The method outperforms prior work across multiple erasure tasks.

**Strengths:**

1. The paper is built on clear observations that different types of concepts emerge at different stages of the denoising process, and addresses this with a principled, targeted erasure strategy.
2. The proposed SSScore + Step Selection mechanism allows the model to automatically identify key timesteps for each concept, improving erasure precision without requiring heuristic or manual selection. Specifically, I praise the usage of the diffusion classifier to guide the training process.
3. The experiments are extensive, covering multiple concept types, metrics (efficacy, specificity, generality), and include ablations, multi-concept erasure, and comparisons with strong baselines.

**Weaknesses:**

1. The SSScore must be computed at every timestep via multiple forward passes using a diffusion classifier. Although this is only done once per concept, the cost is still high, especially when scaling to many concepts or large batches.

2. Although anchor-free erasure is supported, in many cases selecting a suitable anchor is important for quality preservation. The method relies on heuristic selection rules, and the anchor choice significantly affects performance. However, there seems no effective anchor-selection method now (though mentioned in the limitations, but I believe this is really important in this method).

**Questions:**

1. Since the method evaluates SSScore at every single timestep using multiple forward passes, this may become a bottleneck for large-scale or multi-concept erasure tasks. Can this process be approximated or accelerated further?
2. How robust is the method when anchors are suboptimal, or when no clear anchor exists? Can the method adaptively learn anchors instead of relying on heuristic rules?

I expect the authors to response to these concerns. I will raise my rating if they are well-addressed.

---

> ### Author Response · Authors · 2025-11-21
> **Response to Reviewer DjSd Part (1/2)**
>
> Thank you for insightful feedback. We have uploaded a revised version of the manuscript, and all references in this comment correspond to the updated version. We have tried to resolve your concerns as follows:
>
> > **W1:** The SSScore must be computed at every timestep via multiple forward passes using a diffusion classifier. Although this is only done once per concept, the cost is still high, especially when scaling to many concepts or large batches.
> >
> > **Q1:** Since the method evaluates SSScore at every single timestep using multiple forward passes, this may become a bottleneck for large-scale or multi-concept erasure tasks. Can this process be approximated or accelerated further?
>
> Thank you for the valuable concern. However, we would like to clarify that efficiency is not a limiting factor here as SSScore computation is a one-time pre-processing step rather than a repeated cost during training or inference. Moreover, compared with state-of-the-art methods such as ANT, our overall computational cost is reduced by *75%*.
>
> * **Overall Efficiency:** By identifying the critical timesteps, our method avoids fine-tuning across the entire diffusion trajectory. As a result, the total training time is reduced by approximately 75% relative to trajectory-aware baselines like ANT, while also achieving substantially better generation quality. For example, in the celebrity erasure task, the FID improves from 12.47 (ANT) to 7.27 with our method. These results show that the minor overhead introduced by SSScore is far outweighed by the significant gains in both training efficiency and generative performance.
> * **Minimal Overhead:** As detailed in Appendix G.1, for object erasure on CIFAR-10, the entire process takes only about **20 minutes** on a single RTX 4090, with the Step Selection algorithm consuming only less than **3 minutes**.
>
> **Regarding Acceleration (Q1):**
> Thank you for raising this point. You asked if this process can be accelerated. Yes, it can. We explore two strategies to accelerate this process. We have incorporated this discussion in the updated version.
>
> **1. Sparse sampling.** As shown in Figure 6 (Appendix E), the SSScore distribution across timesteps is relatively smooth and continuous. Therefore, we can employ sparse sampling (e.g., computing scores every 5 steps) and interpolate the intermediate values, which would further reduce the pre-calculation cost significantly without compromising accuracy. To this end, we conducted additional experiments to explore the influence of sparse sampling on the model performance. The experimental results are shown in the table below.
>
> | Concept | Method | $CLIP\_{target}\downarrow$ | $FID\_{MSCOCO}\downarrow$ | $CLIP\_{MSCOCO}\uparrow$ | Time cost |
> | :--- | :--- | :---: | :---: | :---: | :--- |
> | **Van Gogh** | original | 21.17 | 7.02 | 31.17 | $161.94s$ |
> | | sparse sampling (3) | 20.43 | 7.62 | 31.12 | $57.17s$ |
> | | sparse sampling (5) | 18.85 | 8.44 | 31.13 | $34.25s$ |
> | **Picasso** | original | 20.98 | 9.06 | 31.13 | $164.62s$ |
> | | sparse sampling (3) | 16.77 | 9.98 | 31.08 | $57.55s$ |
> | | sparse sampling (5) | 18.02 | 9.25 | 31.06 | $33.88s$ |
> | **Monet** | original | 20.78 | 8.09 | 31.17 | $164.02s$ |
> | | sparse sampling (3) | 18.60 | 8.40 | 31.11 | $57.37s$ |
> | | sparse sampling (5) | 20.22 | 7.87 | 31.12 | $33.79s$ |
> | **Average** | original | 20.98 | 8.06 | 31.16 | $163.53s$ |
> | | sparse sampling (3) | 18.60 | 8.67 | 31.10 | $57.36s$ |
> | | sparse sampling (5) | 19.03 | 8.52 | 31.10 | $33.97s$ |
>
> $CLIP\_{target}$ represents the similarity between the target concept and its corresponding prompt, indicating the effectiveness of the erasure.
> $FID\_{coco}$ and $CLIP\_{coco}$ measure the generative performance of the erasure model on the MSCOCO dataset. As shown in the table, sparse sampling performs comparably to the original strategy. It produces slightly stronger erasure and marginally weaker general-purpose generation. Importantly, it substantially reduces the time cost, demonstrating clear practical value.
>
> **2. Concept Clustering.** For large-scale multi-concept erasure, our key observation is that structural concepts primarily emerge in the early–mid denoising stages, whereas fine-grained concepts appear in the mid–late stages. Based on this, we first use an LLM to cluster the concepts to be removed, and concepts within the same cluster require computing the SSScore only once. Specifically, the LLM is prompted to classify each concept into either a *structural* or a *fine-grained* category based on its semantic description, thereby yielding two clusters that correspond to the distinct denoising-stage behaviors of different concept types.
> Table 11 in the appendix is trained using a shared set of SSScore and optimal steps, showing that this approach achieves both efficiency and accuracy in multi-concept erasure.

---

> ### Author Response · Authors · 2025-11-21
> **Response to Reviewer DjSd Part (2/2)**
>
> > **W2:** Although anchor-free erasure is supported, in many cases selecting a suitable anchor is important for quality preservation. The method relies on heuristic selection rules, and the anchor choice significantly affects performance. However, there seems no effective anchor-selection method now.
> >
> > **Q2:** How robust is the method when anchors are suboptimal, or when no clear anchor exists? Can the method adaptively learn anchors instead of relying on heuristic rules?
>
> We appreciate the reviewer’s insightful comment regarding anchor selection. We would like to clarify that SDErasure remains highly robust even without optimal anchors. This is because the anchor serves merely as a tool to improve training stability in a small subset of erasure tasks, rather than a core dependency of the method. Our main paper may not sufficiently elaborate on this aspect, and we have updated the revised version to include a clearer ablation study discussion based on the reviewer’s feedback.
>
> * **The method can operate without relying on heuristic rules.** For fine-grained concepts like celebrities and artistic styles, we actually recommend and use an anchor-free setting (setting the anchor to empty). This is because SSScore naturally identifies mid-to-late denoising stages for these concepts, where structural guidance from an anchor is unnecessary. The state-of-the-art results in Table 2 and Table 3 were achieved using this anchor-free configuration.
> * **The choice of anchor has limited influence on performance.** For structural concepts (e.g., objects), we conducted an ablation study in Table 10 (Appendix H.3) comparing our heuristic anchors against Random (generic concepts like "sky", "forest") and Empty anchors.
>     * Even with Random anchors (Considered as suboptimal anchors), our method achieves an $H_0$ score of **90.21%**, which is superior to all baselines except MACE. It is worth noting that MACE requires images and corresponding masks generated by models such as SAM for training. In contrast, our self-contrastive approach relaxes this requirement and eliminates the need for any image data, substantially improving training efficiency.
>     * Even with Empty anchors, the method maintains effective erasure with an $H_0$ of **88.17%**. This significantly outperforms the baseline ESD, which suffers from low specificity ($Acc_s$ 88.53% vs Empty's 95.84%).
> * **Regarding effective anchor-selection method.** Since our method works robustly with no anchor or randomly chosen anchors, adaptive anchor selection falls outside the scope of our current study. However, we agree that it is a promising direction for future exploration.
>
> This demonstrates that while anchors help, the core performance gains stem from our Step Selection and Quality Regulation ($\mathcal{L}_p, \mathcal{L}_r$), making the method robust to anchor choice.
>
> In addition, prior methods such as UCE, MACE, and RECE also involve the choice of anchors, and different anchors may lead to slight performance variations. Our paper discussed anchor selection as a separate topic, which may have caused some misunderstanding. We clarify that just like previous methods, the performance of SDErasure primarily comes from the core algorithmic design rather than the specific choice of anchors.
>
> **Regarding Learned Anchors (Q2):**
> This is a highly insightful suggestion. Finding suitable anchors for different erasure targets is indeed an interesting and meaningful direction in this research area. While this is not the main focus of our current work, we plan to explore adaptive anchor learning in future research. Thanks again for your insightful comment.

---

> > ### Comment · Reviewer_DjSd · 2025-11-25
> >
> > Thank you again for your response and the improvements made in the revised manuscript. The added experiments and clarifications regarding SSScore efficiency and acceleration strategies (e.g., sparse sampling) are very helpful and appreciated.
> >
> > However, I have two remaining concerns that I hope the authors could clarify:
> >
> > 1. **Diffusion Classifier Behavior When Anchor Concept is Null.** In your method, SSScore relies on a diffusion classifier to compute concept similarity at each timestep. I would like to better understand how this classifier behaves when no anchor concept is provided. My understanding is that diffusion classifiers typically predict among a predefined label space by denoising conditioned on each label and selecting the one with the lowest reconstruction loss.
> > If the anchor concept is empty (i.e., no guiding label or prompt), how is the score computed? How is the classifier guided to measure alignment with the concept to be erased in this case?
> >
> > 2. **Clarification on One-Time Nature of SSScore vs. Use of $\mathcal{S}_t$ During Training**
> > While I agree that sparse sampling reduces the cost and that SSScore computation forms only a small portion of the total erasure pipeline, I would appreciate further clarification on the following point:
> > You state that *“SSScore computation is a one-time pre-processing step”*, but the method uses $\mathcal{S}_t$ (i.e., timestep-specific scores) to select $k$ optimal steps.
> > - Does this mean that you still need to compute the SSScore at $k$ different timesteps during the actual erasure training? If so, the process seems to involve more than a single evaluation and is used iteratively across selected steps.
> > - Moreover, based on my understanding, the diffusion classifier's ability to distinguish concepts may evolve as training progresses. Could this affect the relevance of the pre-computed $\mathcal{S}_t$ values? Or is the assumption that the classifier’s performance remains stable during erasure?
> > Additionally, could you confirm the computational complexity of computing a single $\mathcal{S}_t$? Is it $O(m \cdot 2T)$, where:
> > - $m$ is the number of concepts to erase, and
> > - $T$ is the number of diffusion steps (assuming one target concept + one anchor concept)?

---

> > > ### Author Response · Authors · 2025-11-26
> > > **Response to Reviewer DjSd Part (1/2)**
> > >
> > > Thank you for insightful feedback. We are happy to provide further clarifications on the technical details.
> > >
> > > > **Q1:** Diffusion Classifier Behavior When Anchor Concept is Null.
> > >
> > > This is an excellent question regarding the mechanics of the "Anchor-Free" setting. We clarify that in text-to-image diffusion models, an empty anchor does not imply a lack of input. Instead, it corresponds to "unconditional conditioning". This design is closely related to the idea behind classifier-free guidance (CFG), where unconditional embeddings serve as a baseline signal rather than a genuinely defined Null in implementation.
> > >
> > > **1. Theoretical Basis: Unconditional Conditioning**
> > >
> > >  During the training of diffusion models, text conditions are randomly dropped with a certain probability to enable Classifier-Free Guidance. This training strategy ensures that the model learns to generate meaningful, high-quality natural images even without specific text prompts. Therefore, the model's prediction under "unconditional input" represents its inherent understanding of the general natural image distribution (the prior), unconstrained by specific semantic directives.
> > >
> > > **2. Mechanism of SSScore in Anchor-Free Mode**
> > >
> > > When the anchor is set to empty, the SSScore compares the **target concept's trajectory** against this **general natural image trajectory**:
> > >
> > > $$
> > > \mathcal{S}\_t \approx \frac{p(c\_{\text{target}} | x\_t)}{p(c\_{\text{target}} | x\_t) + p(c\_{\text{unconditional}} | x\_t)}
> > > $$
> > >
> > > This comparison acts as a "Concept Emergence Detector":
> > >
> > > * High $\mathcal{S}\_t$: A high score indicates that the conditional prediction (with the target prompt) is significantly more accurate than the unconditional prediction. This implies that the model is currently generating concept-specific features that cannot be explained by the general image prior. This marks a critical divergence point where the concept is actively shaping the generation.
> > >
> > > * Low $\mathcal{S}\_t$: A low score indicates that the target prompt provides little information gain over the empty prompt. This typically occurs in steps where the model is generating generic features that are shared across the natural image manifold and are not specific to the target concept.

---

> > > > ### Author Response · Authors · 2025-11-26
> > > > **Response to Reviewer DjSd Part (2/2)**
> > > >
> > > > > **Q2:** Clarification on One-Time Nature of SSScore vs. Use of $\mathcal{S}\_t$ During Training.
> > > >
> > > > We confirm that the SSScore is computed **once** using the original, frozen model before any erasure training begins. It is not re-evaluated or updated during the fine-tuning process. Although the preprocessing involves calculations across multiple timesteps, this procedure is performed entirely prior to erasure training and thus incurs only a modest computational cost.
> > > >
> > > > * **Implementation:** We identify the set of critical timesteps $\mathcal{T}_{\text{critical}} = \lbrace\ t \mid \mathcal{S}_t > \lambda \rbrace$ during pre-processing. During the actual erasure training, we simply sample timesteps from the distribution and apply the erasure loss. No forward passes for classification are performed during the training loop.
> > > >
> > > > * **Why one-time preprocess?** We utilize the SSScore derived from the frozen pre-trained model as a fixed reference for intervention. Since the objective of concept erasure is to minimize the divergence between the target and anchor distributions, successful training does **affect the relevance of the pre-computed $\mathcal{S}\_t$**. If the SSScore were re-evaluated during training, this convergence would result in posterior collapse, yielding an uninformative uniform distribution across timesteps. For instance, in celebrity erasure, successful suppression at the initially identified optimal steps diminishes their SSScores. A dynamic update might consequently misidentify suboptimal early or late stages as the new critical intervals, leading to the structural degradation or incomplete erasure illustrated in Figure 1. Thus, a static evaluation is necessary to maintain a stable objective for identifying and suppressing critical generative phases.
> > > >
> > > > > **Q3:** Computational complexity of computing a single $\mathcal{S}\_t$.
> > > >
> > > > **1. Clarification on Single Score Complexity:**
> > > > To be precise, the complexity of computing a **single** $\mathcal{S}\_t$ (at a specific timestep $t$ for one concept pair) is independent of the total number of concepts ($m$) or the total diffusion steps ($T$). It depends only on the number of noise sampling iterations ($K$) used to estimate the expectation. Since $K$ is a small constant (the default value is 20), the unit cost is **$O(1)$**.
> > > >
> > > > **2. Confirmation of Total Complexity:**
> > > > You are correct regarding the aggregate cost. The formula you proposed, $O(m \times 2T)$ accurately describes the computational complexity for the **entire pre-processing phase** to generate full SSScores for all targets. As validated in our rebuttal, applying sparse sampling significantly reduces the effective $T$, making this one-time pre-processing highly efficient.
> > > >
> > > > We sincerely thank you again for your constructive feedback, which has helped us significantly improve the clarity and completeness of our work. We will ensure that these explanations regarding the anchor-free mechanism and computational complexity are incorporated into the final manuscript.

---

> > > > > ### Comment · Reviewer_DjSd · 2025-11-26
> > > > >
> > > > > Thank you for the detailed explanation regarding the one-time computation of SSScore and its usage during erasure training. I now understand that this is effectively a **step selection mechanism** that identifies critical timesteps for intervention, based on the semantic divergence between the target and anchor concepts. The rationale for using a frozen model and avoiding dynamic re-evaluation is also well-explained.
> > > > >
> > > > > However, I still have one conceptual question regarding the **motivation and design** of the SSScore-based scheduling.
> > > > >
> > > > > It is well-established in prior diffusion model literature that **early denoising steps primarily capture coarse structure**, while **later steps handle fine-grained details**. Based on this, one might ask:
> > > > >
> > > > > > Could a simple, naive schedule, such as a linearly increasing weight over time, or even a binary mask that applies erasure loss only after a certain timestep, already suffice to achieve the desired erasure effect?
> > > > >
> > > > > Such naive schedules would be far less computationally expensive, and might still align well with the observed semantic progression of diffusion sampling.
> > > > >
> > > > > - Have you considered or experimented with such baseline step selection strategies (e.g., linear scheduling or binary timestep masks)?
> > > > > - If such experiments have been conducted, it would be very helpful to see a comparison between these naive baselines and your SSScore-based selection in terms of erasure effectiveness and generation quality.
> > > > > - If such experiments were not conducted, that's ok. Just explain that, either heuristically or mathematically (better), on the specific advantages your method offers over these simpler heuristics?

---

> ### Author Response · Authors · 2025-11-28
> **Response to Reviewer DjSd**
>
> We thank the reviewer for this thoughtful conceptual question regarding the necessity of SSScore compared to simpler heuristics.
>
> In our initial exploration, we investigated simple strategies such as manual timestep selection. However, we found these approaches to be ineffective and heavily reliant on manual heuristics.
> To empirically demonstrate these limitations, we conducted an additional experiment comparing SDErasure against a **Linear Weighting Schedule**. We will update the manuscript with these results.
>
> We evaluated the erasure of both Artistic Styles (Fine-grained) and Objects (Structural) using both the Naive Linear Schedule and our SSScore-based method. The results reveal that a static schedule fails to adapt, leading to opposite failure modes.
>
> **Table A: Erasure of Artistic Styles (Fine-grained).**
> Metric: CLIP (Lower = Stronger Erasure), FID (Lower = Better Quality)
>
> | Concept | Method | $CLIP \downarrow$ | $FID \downarrow$ | Analysis |
> | :--- | :--- | :---: | :---: | :--- |
> | Van Gogh | Linear | 17.52 | 10.91 | **Over-Erasure** |
> | | Ours | 21.17 | 7.02 | Balanced |
> | Picasso | Linear | 16.90 | 11.43 | **Over-Erasure** |
> | | Ours | 20.98 | 9.06 | Balanced |
> | Monet | Linear | 15.86 | 12.03 | **Over-Erasure** |
> | | Ours | 20.78 | 8.09 | Balanced |
>
> **Table B: Erasure of Objects (Structural).**
> Metric: $Acc_e$ & $Acc_g$ (Lower = Better Erasure/Generality)
>
> | Concept | Method | $Acc_e \downarrow$ | $Acc_g \downarrow$ | Analysis |
> | :--- | :--- | :---: | :---: | :--- |
> | Airplane | Linear | 57.47 | 73.87 | **Under-Erasure** |
> | | Ours | 4.37 | 2.66 | Successful Erasure |
> | Automobile| Linear | 16.32 | 24.95 | **Under-Erasure** |
> | | Ours | 2.77 | 11.54 | Successful Erasure |
>
> The results reveal a clear contrast: static linear scheduling leads to over-erasure for fine-grained styles (indicated by high FID) while causing under-erasure for structural objects (indicated by high $Acc_e$ and $Acc_g$). In contrast, our SSScore-based method achieves balanced performance across both tasks, demonstrating its adaptability and effectiveness.
>
> We attribute these failures to the inherent lack of adaptivity in static linear weighting, which cannot adjust the erasure phase according to specific concepts. For object erasure, effective removal requires targeting holistic structural features, necessitating intervention in the early-to-mid denoising stages. Conversely, applying this same erasure during early stage to fine-grained artistic styles results in over-erasure, as these tasks require preserving structural details while only removing texture. Our SSScore-based method addresses this limitation by leveraging the discriminative capability of the Diffusion Classifier to pinpoint concept-specific features. This allows for the **adaptive** adjustment of the erasure phase, thereby achieving superior performance.
>
> Evaluation of **Binary Timestep Masks (Figure 1)**.
> Our main paper's Figure 1 serves as an empirical evaluation of this strategy. The comparison of "Early", "Middle", and "Late" erasure is effectively equivalent to applying fixed binary masks. The results demonstrate that static masks lack generalizability: applying a "Late" mask to "Cat" causes under-erasure, while an "Early" mask causes over-erasure. This confirms that no single binary mask works across diverse concepts, necessitating our adaptive approach.
>
> This further supports our conclusion:
> fixed timestep strategies (binary or linear) lack generality and fail to adapt to concept-specific behavior.
>
> Thank you again for this constructive suggestion, which has helped us significantly strengthen the empirical verification of our method's necessity. We hope these new results and analyses fully resolve your concerns.

---

### Comment · Area_Chair_MKvz · 2025-11-22

Dear Reviewers,

Thank you for your time and effort in reviewing submissions for ICLR  2026. As we begin the author-reviewer discussion process, we kindly remind you to submit your responses to the author rebuttals by **December  2**.


Your engagement in this discussion phase is crucial to ensuring a fair and thorough evaluation of each submission.

**Action Required**


- Carefully consider the authors’ rebuttal and any additional evidence they provide.

- Update your review (if applicable) to reflect your revised perspective.

-  **Discuss with the authors if further details are required**


Your AC

---

### Author Response · Authors · 2025-12-02
**Review and Reviewer-Author Discussion Summary**

Dear Area Chair and Reviewers,

As the active discussion phase had to be closed due to the privacy-leak, we would like to provide a concise summary of our rebuttal to assist the Area Chair in their final assessment.

First and foremost, we sincerely thank all reviewers for their constructive feedback. We value these insights highly and have incorporated corresponding revisions and improvements into our manuscript. We would like to express our particular gratitude to Reviewers **DjSd** and **DdAx**. Reviewer DjSd engaged in a productive multi-round discussion with us, raising many valuable questions. Although the discussion window closed before further interaction could take place, we are confident that our responses have fully addressed their concerns. We also thank Reviewer DdAx for acknowledging the strength of our experimental results and analysis, and for **raising their score**. We deeply appreciate this endorsement.

We further believe that we have effectively addressed all concerns raised during the review process, significantly strengthening the paper in the process.

**Summary of Key Rebuttal Contributions**

* **Theoretical Grounding & Optimality Verification (Re: drbX, DdAx):** We provided a formal derivation linking SSScore to the instantaneous posterior probability, establishing a solid theoretical foundation. We further validated this via a new Group Ablation Study, confirming that the identified timesteps theoretically and empirically yield the optimal erasure performance. We have included detailed derivations and experimental results in the **Method section** and **Appendix E.**
* **Computational Efficiency & Acceleration (Re: DjSd, tk2R):** We clarified that SSScore is a one-time pre-processing step and introduced a Sparse Sampling strategy. This reduces pre-computation cost by **~80%** without compromising accuracy. Detailed analysis and experiments are provided in **Appendix C.**
* **Robustness of Anchor Selection (Re: DjSd, drbX):** We conducted extensive ablations demonstrating that SDErasure is not dependent on optimal anchors. Even with "Random" or "Empty" (anchor-free) settings, our method significantly outperforms baselines, proving that robustness stems from the core Step Selection mechanism. We have incorporated these detailed experimental results into the **Ablation Study section.**
* **Necessity of Adaptivity (vs. Static Schedules) (Re: DjSd):** We compared SDErasure against a linear weighting schedule, revealing that static schedules suffer from a dual failure mode: over-erasure for fine-grained styles and under-erasure for structural objects. This empirically proves the necessity of our concept-specific adaptive strategy.
* **Scalability in Multi-Concept Erasure (Re: tk2R, DdAx):** We added a direct comparison against strong baselines (UCE, MACE, ANT) on the challenging task of erasing multiple concepts simultaneously. SDErasure achieves superior fidelity compared to baselines that suffer from significant degradation, establishing a new state-of-the-art in large-scale erasure. These results have been included in **Appendix I.3.**

**Conclusion**

The reviewers collectively highlighted the significance of SDErasure, praising the "adaptive timestep selection strategy" as "novel and well-motivated" (DdAx) based on the "key observation" that only specific timesteps are critical (drbX). They commended the "principled, targeted erasure strategy" (DjSd), specifically noting that the SSScore mechanism improves "erasure precision without requiring heuristic or manual selection" (DjSd). Furthermore, the proposed Quality Regulation was recognized as "an important contribution" (drbX) that "mitigates the instability issues of fine-tuning" (tk2R). The consensus is that our framework achieves a "strong balance between effective concept erasure and preserving generative quality" (drbX), demonstrating "superior performance" (tk2R) through "extensive" (DjSd) experiments.

SDErasure introduces the first targeted erasure framework based on concept-specific generative phases. We have substantiated this approach with rigorous theoretical derivations and extensive empirical analysis to validate the effectiveness of the SSScore mechanism. Consequently, SDErasure achieves State-of-the-Art performance (reducing FID from 9.51 to 6.74), effectively eliminating target concepts while maximizing the preservation of unrelated generative capabilities. We believe the comprehensive rebuttal and added experiments have fully resolved all concerns and significantly enhanced the completeness and robustness of this work.

Thank you again for your time and consideration.

Best regards,

The Authors

---

### Meta-Review · Area_Chair_pzUr · 2026-01-07

**Summary:**

The initial review scores for this paper are 6 (reviewer DjSd), 6 (reviewer tk2R), 4 (reviewer drbX), 4 (reviewer DdAx).

And the major concerns from reviewers are:
--lacks an ablation study on the timestep selection strategy.

--generalizability of the step selection strategy

--more analysis on SSScore

--established methods in multi-concept erasure experiments

--sensitive to anchor concept selection

--theoretical justification for why the selected timesteps are optimal

--performance gains are inconsistent

--related works are missing

--threshold in the Step Selection algorithm

--clarity of some technical parts

**Reviewer Concerns:**

Most concerns seems to be addressed, probably except "threshold in the Step Selection algorithm".

**Reviewer Scores:**

DdAx already mentioned they will raise the score to 6. drbX might also increase the score, since most concerns are addressed.

---

### Decision · Program_Chairs · 2026-01-26

Accept (Poster)